# Open High-Resolution Satellite Imagery:
# The WorldStrat Dataset – With Application to Super-Resolution

**Julien Cornebise**[*]
University College London
& Why How Ltd
j.cornebise@ucl.ac.uk

**Ivan Oršolić**[*]
Why How Ltd
ivanorsolic@gmail.com

**Freddie Kalaitzis**
University of Oxford
freddie.kalaitzis@cs.ox.ac.uk

## Abstract

Analyzing the planet at scale with satellite imagery and machine learning is a dream that has been constantly hindered by the cost of difficult-to-access highly-representative high-resolution imagery. To remediate this, we introduce here the **WorldStrat**ified dataset. The largest and most varied such publicly available dataset, at Airbus SPOT 6/7 satellites' high resolution of up to 1.5 m/pixel, empowered by European Space Agency's (ESA) Phi-Lab as part of the ESA-funded QueryPlanet project, we curate nearly 10,000 km² of unique locations to ensure stratified representation of all types of land-use across the world: from agriculture to ice caps, from forests to multiple urbanization densities. We also enrich those with locations typically under-represented in ML datasets: sites of humanitarian interest, illegal mining sites, and settlements of persons at risk. We temporally-match each high-resolution image with multiple low-resolution images from the freely accessible lower-resolution Sentinel-2 (S2) satellites at 10 m/pixel. We accompany this dataset with an open-source Python package to: rebuild or extend the WorldStrat dataset, train and infer baseline algorithms, and learn with abundant tutorials, all compatible with the popular eo-learn toolbox. We hereby hope to foster broad-spectrum applications of ML to satellite imagery, and possibly develop from free public low-resolution Sentinel-2 imagery the same power of analysis allowed by costly private high-resolution imagery. We illustrate this specific point by training and releasing several highly compute-efficient baselines on the task of Multi-Frame Super-Resolution. License-wise, the high-resolution Airbus imagery is CC-BY-NC, while the labels, Sentinel-2 imagery, and trained weights are under CC-BY, and the source code and pre-trained models under BSD, to allow for the widest use and dissemination. The dataset is available at https://zenodo.org/record/6810792, the software package at https://github.com/worldstrat/worldstrat, and homepage at https://worldstrat.github.io.

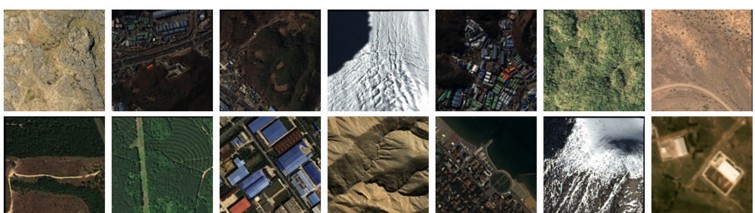

Figure 1: A glimpse at the variety of land uses covered by high-resolution imagery in the dataset.

---

[*]Equal contribution

36th Conference on Neural Information Processing Systems (NeurIPS 2022) Track on Datasets and Benchmarks.

# 1 Introduction

## 1.1 The Problem

Computer vision and satellite imagery seem to be a match made in heaven. The idea to automatically process the growing amount of imagery collected has been lingering for decades in the remote sensing and the earth observation communities. The appeal of seeing the whole planet, and analyzing it at scale, is akin to few others. Many attempts have been made throughout the last thirty years. The ever-higher resolution of imagery available to civilians, and the last decade of improvement in Machine Learning and Computer Vision, have brought tools that could be brilliantly assistive in that regard. Some very visible scientific successes have been published, such as (Jean et al., 2016). Even prominent Deep Learning innovators cut their teeth early on as Master students on applications to satellite imagery, e.g. Mnih and Hinton (2010). And prominent tools like Google Earth, which gives everyone access to high-resolution aerial imagery (although not all of it obtained by satellite), is a trigger for imagination. The dream runs deep.

However, the full potential of these two fields has been hindered by a combination of factors, in particular data access and the associated costs.

Broadening access to cutting edge technology is a hacker's delight: some satellite imagery can actually be received by any accessible for any amateur with an antenna up their roof – back as an undergrad the first author built a cheap reception, storage, and processing station for low-resolution Meteosat Second Generation imagery (Beaudoin et al., 2005), undercutting ten-fold the cost of commercial stations at the time. And while such refinements on reception stations are not feasible for high-accuracy digital imagery, since 2015, the European Sentinel-2 (S2) satellites have been providing medium-resolution imagery (10 m/pixel) for free every five days accross the world for anyone who knows how to access them.

Yet, high-resolution imagery (1 m/pixel) or very-high-resolution (sub 1 m/pixel) are still out of easy reach. In a more recent work with Amnesty International to detect destroyed villages in conflict zones, (Cornebise et al., 2018), we discovered that the cost to purchase a single very-high-resolution mosaic of the whole of Darfur, akin to Google Earth at maximum zoom, would cost 4 Million USD, even including a generous discount for charities. This makes it a tremendous challenge to even experiment with computer vision for high resolution.

Even setting cost aside, and assuming, as some hope, that the thunderous technological advances in launch technologies unlock a deluge a high-resolution imagery at a smaller price point, the key material for Machine Learning is still simply not there: carefully curated datasets to train on! Accessing satellite imagery, even Sentinel 2, requires a certain amount of domain knowledge, more so than for natural images that can be sourced pretty much anywhere. The barrier to entry is real.

Indeed open high-resolution satellite imagery datasets are still quite rare, and the few who do exist tend to be either small, cover few unique locations, ad-hoc locations, or are designed for very specific uses. The SpaceNet challenge datasets (Van Etten et al., 2018) are widely used satellite imagery datasets. Their combined unique location area is close to WorldStrat with a bit more than 10 000 km², but it is mainly focused on urban structures and made for specific tasks like building or road detection, with varying data providers and resolutions, and with no paired multi-temporal low-resolution imagery.

We mention paired lower-resolution, because another hope to work around the lack of access lies in the field of Super-Resolution: being able to derive from (possibly multiple) free low-resolution satellite revisits the same insights that would be available from a single high-resolution satellite visit of the corresponding area. While we want to build datasets that can be used for a whole breadth of applications, making them suitable for super-resolution brings a swath of extra benefits.

On that topic, the ESA Kelvins PROBA-V dataset (Märtens et al., 2019) and the associated competition have been a boon for such multi-frame super-resolution, but is single-channel and most importantly is very low resolution at 300 m/pixel and 100 m/pixel. It is also not georeferenced or time-referenced. However, it allowed us to develop the high-performing HighRes-Net multi-frame super-resolution algorithm (Deudon et al., 2020). There is a an abundant literature on multiframe super-resolution for natural images and videos, but quite not as much in satellite imagery – which we will cover in Section 4.1.

The very recent dataset by Michel et al. (2022) promises to enable single-image super-resolution. covers only 29 unique locations, on 806 unique km² , at 5 meter/pixel. It offers nine revisits for each location, remarkably pairing one single low-resolution and one high-resolution for each visit. This is somewhat promising, but still lacks the breadth we hope for.

To the best of our knowledge, no dataset tries to be generic and cover systematically the whole type of land-use across the world. Even fewer are explicitly designed with the aim of transferring learning to high-availability lower-resolution data: low-resolutions from Sentinel 2 can be manually added, but at the price of extra work and expertise.

## 1.2 Our Contributions

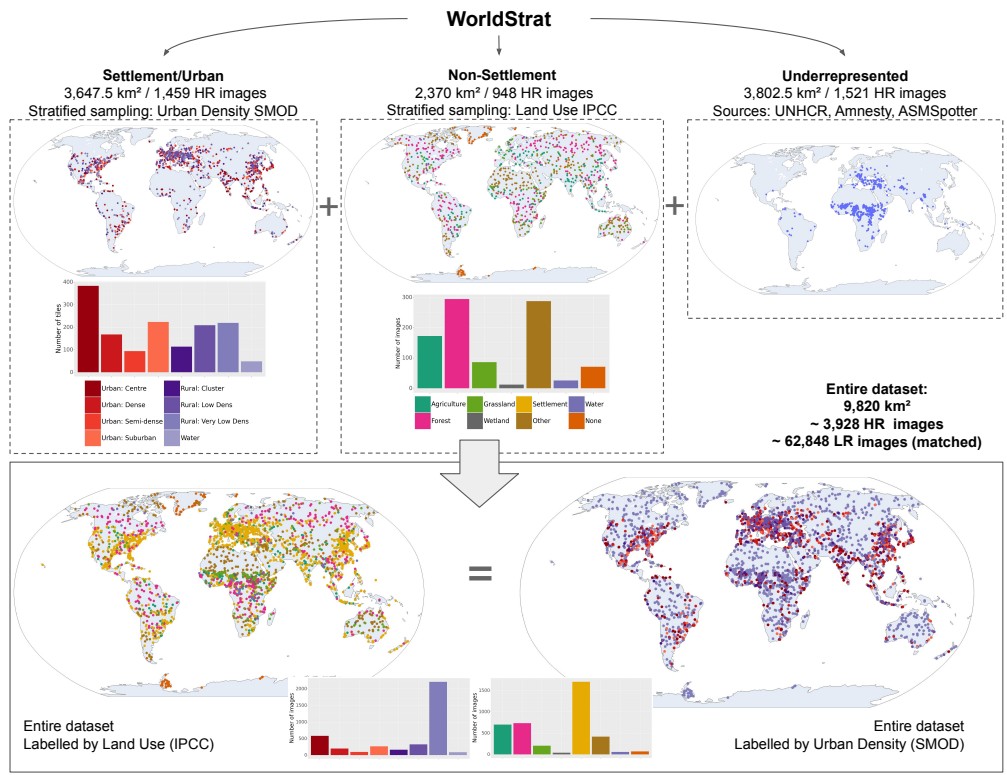

Figure 2: Summarizing the construction and classes of the WorldStrat dataset.

We aim to empower the development of machine learning for satellite imagery that can be used for a wide range of applications: from ecology and climate change monitoring, to urbanization, to sociology, to disaster preparation, to agriculture, etc. Our focus while building this dataset and its accompanying benchmarks and software package has therefore been to:

- Maximise the representation of all possible features of interest, for the widest possible use-cases.
- Have a decent worldwide representativity – especially relevant in light of the need for Fairness Accountability and Transparency in ML, which includes the problem datasets often biased towards the Global North.
- And provide a pipeline that allows easy reproducibility by others, and extension if extra budget becomes available.

Our resulting dataset, the **World Stratified Dataset** (or WorldStrat for short) (Cornebise et al., 2022) covers ∼**10,000 km² , and 3504 distinct locations, specially curated for the highest diversity of possible uses**. In particular, as visualised in Figure 2, we separated our image acquisition budget into three parts:

- One part focusing on human settlements further stratified by population density: filtering the world for settlements according to the European Space Agency's (ESA) Climate Change Initiative (CCI) Land Cover product (ESA, 2017) (which is licensed without restrictions under the CCI Data Policy v1.1), then sub-stratifying according to the Urban density class of the Global Human Settlement Layer SMOD product (Florczyk et al., 2019) (which is licensed under CC-BY).
- A second part focused on non-settlement areas, using stratified sampling and class-rebalancing across the Land Cover Classification System labels of the ESA CCI Land Cover product, and more precisely its aggregation into classes provided by the International Panel on Climate Change (IPCC).
- Finally, a third part is focused on use cases typically under-served by usual datasets and not covered by the above. We sourced points of interests from the United Nations High Commissioner for Refugees (UNHCR) for populations of concerns (which is licensed under CC-BY-IGO), from Amnesty International for human rights sites of interest (with permission), and ESA artisanal and small-scale mining (ASM) ASMSpotter for illegal mining (with permission).

At each resulting location, along with the label, we provide the following imagery:

- One High-Resolution multispectral image from Airbus SPOT 6/7 (licensed a paid-for extension to ESA's Third Party Missions (TPM) license granting redistribution under CC-BY-NC), in RGB (6 m/pixel), Near Infrared (6 m/pixel), and Pan-chromatic channels (1.5m / pixel), at 1054x1054 pixels at the highest resolution.
- 16 Low-Resolution revisits from Copernicus Sentinel-2 (licensed without restrictions under Copernicus Sentinel Data Legal Notice and Service Information), temporally matched to the High-Resolution image – within 5 days for the temporally closest. All 13 spectral bands are covered, at up to 10 m/pixel.

The rest of the article is structured as follows. In Section 2, we present how we have curated the parts of the world we cover, aka Areas Of Interest (AOIs), to offer maximum representativity of the world and of use cases. In Section 3, we describe the characteristics of the paired imagery available at every AOI, both in low- and high-resolution, and how it can be easily extended. In Section 4.1, to illustrate one possible use of this dataset, we establish baselines on multi-frame super-resolution tasks using several architectures, with an emphasis on compute efficiency. We also present the toolbox integrated with popular package eo-learn to use, reproduce, and extend, all our work. We conclude in Section 5 by discussing ideas tried and discarded along the way, as well as possible extensions.

## 2 Curating Highly Representative Locations

The WorldStrat dataset covers almost 10,000 km² . Each base AOI is 2.5 km² , i.e. 1,581 meters per side, the minimum contiguous order size allowed by Airbus, provider of the high-resolutiom imagery. This maximizes the number of AOIs within the allotted budget.

### 2.1 Stratifying The World

We use the first half of the dataset to attempt a systematic, stratified coverage of the world. The question becomes: how do we chose these locations to ensure a "best" application-agnostic dataset for super-resolution?

Sixty precent will be taken from the "Settlement" class from the ESA CCI LandCover Product, which we then stratify according to the Global Human Settlement Layer (GHSL) Settlement Grid (SMOD) for different types of urban density, and with marginal distribution proportional to the cubic root of the actual distribution -- to keep the order of classes but diminish the overall imbalance.

Fourty percent will be taken from all the other IPCC classes, i.e. non-settlement, stratified according to (non-settlement) IPCC class, marginal distribution proportional to the cubic root of the actual distribution, and within each (non-settlement) IPCC class, again stratifying, according to the LCCS class (thinner vegetation typology), again with cubic root proportions.

## 2.2 Why stratifying by land-use

In an optimal sampling entirely focused on super-resolution, we would first find a latent-space feature-based clustering of satellite imagery, not unlike (Jean et al., 2019). Then, inspired by the classical statistical tools variance-minimizing design of experiments (Atkinson et al., 2007), we would sample each semantic cluster proportionally to how difficult it is to super-resolve.

Quantifying this would, at best, be done via some form of active learning, with the problem of being dependent on the actual super-resolution algorithm used in the process, or at worst, require having first solved the super-resolution problem, neither of which is quite satisfying. Besides, this would, again, be optimized solely for super-resolution: we aim for broader.

A proxy frequently used in statistical design of experiment is instead be to sample each semantic class proportionally to its variance. While this works well for numerical features, variance is not a clearly defined quantity for visual features. There would be an interesting methodological question to be investigated as to what a form of "semantic variance" would look like, but is quite a deep methodological topic and would take us into a rabbit hole out of scope of this project.

This ideal schema being quite ill-defined, we move on to a more pragmatic choice.

A pragmatic way to decide on an optimal dataset suitable for a broad range of applications is instead to ensure a representation of every type of land. This prepares equally for studies of urban space, of ice coverage, of limnology, etc.

We could sample POIs uniformly on POIs uniformly on the World Geodetic System 1984 (WSG84) ellipsoid, conditionally on being on land (by, say, rejection sampling), then filter the resulting points of interest (POIs) by SPOT availability. However this would not provide any guarantee as to how much each type of land use would be represented, nor in which proportion. The law of large numbers would guarantee a representation asymptotically proportional to the real world, but this does not provide any fine-grained control, nor any representation of rare classes: random fluctuations would not be controlled for, which is problematic when the number of samples is "far from asymptotic" – as in our case.

Instead, we can stratify by "land use", for some definition thereof, exploiting ample prior art on land use and land cover classifications.

The ESA Climate Change Initiative Land Cover Products (ESA CCI LCP) (ESA, 2017). This worldwide land cover map is based on Food and Agriculture Organization of the United Nations (FAO) Land Cover Classification System (LCCS) hierarchical system (Di Gregorio, 2005). Moreover, it nests it hierarchy within a coarser classification suggested by the International Panel for Climate Change ESA (2017)[page 30.], reflecting the preoccupations of climate change studies.

### 2.2.1 Sampling the Non-Urban World

We sampled 2,000 km² at 800 POIs and stratified them using the IPCC classification found in the ESA CCI Land Cover 2019 dataset. We then rebalanced the classes using cubic-root importance sampling, e.g. to under-represent on-purpose the surprisingly large planet-wide proportion of moss/lichen-covered land. The points were enriched to include ice-caps, noticeably absent from the original classification but of core importance both for climate analysis, geopolitical implications, and new logistics routes.

The rationale for using importance sampling with the cubic root of the population distribution as the proposal is the following:

- We want to make sure we have a higher representation of rare classes, i.e. not being dominated by the classes that are already highly present naturally, because we wouldn't want our super-resolution to miss a rare object that would be out of place.
- But we do still want to acknowledge that classes that are highly present naturally are possibly be super-resolved quite often, so we do not want to completely flatten the histogram by sampling uniformly.

Hence the cubic root as a compromise, which boosts rare classes but does keep the monotonous order relationship between the classes. The result is visible in Figure 3.

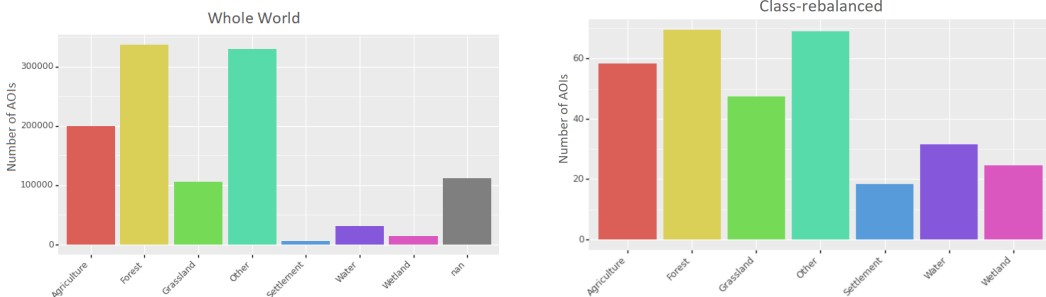

Figure 3: Distribution of the IPCC classes over the whole world (left), and the class-rebalanced sample using cubic root (right).

### 2.2.2 Sampling the Urban World: Nested Stratification of Urban Density with GHSL SMOD

We sampled 3,000 km² at 1,200 POIs and filtered them by the "Urban" class found in the ESA CCI dataset. The points were then stratified by the density in Global Human Settlement Layer SMOD, which combines census and build-up data. The SMOD classes were also rebalanced by cubic-root importance sampling. We wanted to oversample the Settlement class, as it is quite crucial for a very wide range of applications, from geography to economics to demographics to aid planning to disaster recovery.

The Global Human Settlement Layer, an in particular its SMOD product (Florczyk et al., 2019), focuses on density of building and population. A titanic work, it is not without shortcomings, documented by van Den Hoek and Friedrich (2021), especially regarding informal settlements. Those would make it ill-suited for our needs, were it not complemented by the UNHCR-provided POIs.

Within the 1200 POIs classified as Settlement under the IPCC classification in ESA CCI LC, we stratify with respect to GHSL SMOD. ESA CCI LC is of higher spatial resolution than SMOD. But IPCC is of coarser semantic resolution than SMOD. SMOD has indeed multiple settlement classes, visible in Figure 2.

The relation in spatial resolution is inverse of the relation in land use resolution. Therefore it makes extra sense to stratify IPCC settlement tiles (which are of a smaller size than SMOD tiles) according to the type of (larger) SMOD tile they fall into. This guarantees that we will have sampled "IPCC settlements tiles" from a varying type of surrounding urban environment (as provided by SMOD).

### 2.3 AOIs from Under-Represented Key Users

The second half of the dataset is obtained by sourcing 3,895 sq km around 1,062 Points Of Interest (POIs) from specialists of use-cases ignored by most existing datasets. For the rarer type of POIs, we sample 9 actual images in a non-overlapping grid centered on the POI.

Non-governmental organizations (NGOs) and charities in Earth Observation (EO) interested about using recent Machine Learning articles are often met with the sobering answer "Sorry, our models have never been trained on this type of landscape". This is for example the case for informal settlements, refugee camps. Most existing satellite imagery datasets, such as SpaceNet (Van Etten et al., 2018), are often centred on urban features in large cities – when they even cover beyond North America.

We also know that some of the highest benefits of this technique will go to social change actors who otherwise absolutely cannot afford high-resolution imagery.

For that reason, we contacted high-potential users to inquire about the type of landscape they operate in. By obtaining examples of their past and ongoing points of interest (POIs) we ensure that our dataset will contain examples of the type of terrains in regions typically relevant to those users.

**Amnesty International**, focusing on human rights violations and conflict areas: 495 km² at 22 POIs, with 9 AOIs around each POIs. Amnesty International's geospatial specialist Micah Farfour, specialised in satellite monitoring of human rights abuse and environmental abuse, provided us with

22 POIs across the globe, ranging from barracks to prisons to mass grave sites. To make up for their small number, we ordered each of them on a 3x3 grid of minimum-size tiles, i.e. a total of 22.5 km² per POI, with a total of 495 km² .

**ASMSpotter**, focusing on illegal mining in remote areas. 900 km² at 40 POIs, with 9 AOIs around each POIs, mostly in South America. ASMSpotter's data specialist Moritz Besser provided 40 POIs which we ordered similarly to Amnesty's, covering 900 km² .

**United Nations High Commissioner for Refugees (UNHCR)**, focusing on informal human settlements in disaster and conflict areas. 2,500 km² at 1,000 POIs, one AOI per POI. Oregon State University professor Jamon Van de Hoek, specialised in human settlements and in particular informal settlements – see e.g. his latest publication van Den Hoek and Friedrich (2021) studying the Global Human Settlement Layer – kindly provided us with 3,000 POIs from the Persons of Concerns UNHCR Dataset (UNHCR, 2021), which we downsampled to 1,000.

Since certain parts of the world have a particularly high density of UNHCR-register location (e.g. in the Middle East), we avoided overlapping AOIs by ensuring all our AOIs are at least 10 km apart, using rejection sampling.

## 3 Imagery

### 3.1 High-Resolution: Single Visit SPOT 6/7

Each AOI has a single visit of SPOT 6/7 high-resolution imagery, over five spectral bands. The panchromatic (PAN) band has a resolution of 1.5 m/pixel, hence a 1,054x1,054 pixel image per 2.5 km² AOI. The Red, Green, Blue, and Near Infrared bands (RGBNIR) are each at 6 m/pixel. The date of the visit has been picked at random between 2017 and 2019 amongst the visits whose whole-scene cloud-cover is lower than 5%. Because our AOIs are much smaller than a full SPOT scene, it is not absolutely guaranteed that the actual image has precisely 5% cloud – it is likely to be entirely empty of clouds. This provides a good target image to reconstruct in the case of super-resolution. A glimpse of the imagery is show in Figure 1.

### 3.2 Low-Resolution: Multiple Revisits Sentinel 2

For each high-resolution visit, we have 16 revisits. We picked revisits' dates centered on SPOT visit. If more than 16 revisits per SPOT visit are needed, they are available via SentinelHub. The average time between revisits is 5 days.

The low-resolution revisits are provided in each of two product types: Level-1C (L1C) which provides top-of-atmosphere (TOA) reflectance images in cartographic geometry; Level-2A (L2A) which provides bottom-of-atmosphere (BOA) reflectance images derived from the associated Level-1C products by the European Space Agency. For each revisit, 13 bands are available in the L1C product type, and 12 bands in the L2A product type. The resolution ranges from from 10 m/pixel (for RGB) to 60 m/pixel.

We chose to not filter the low resolution Sentinel-2 revisits by their cloud coverage. This is to try and ensure the training distribution on the low resolution is similar to the real world use cases, where the user will want to rebuild at a given place at a given time. Algorithms should learn to ignore clouds and be able to assemble a view from the cloudless parts of the cloudy revisits.

### 3.3 Temporal Selection: Matching AOI, Low-Res, and High-Res Imagery

Unlike Sentinel-2, SPOT is only available where and when it has been tasked. This raises two questions: How available and at which dates is SPOT high-resolution imagery of the POIs we have sampled? For each SPOT visit, how available is the Sentinel-2 imagery, within which time window around the date of the SPOT visit?

Figure 4 illustrates the number of Sentinel-2 revisits (Y-axis) over each start and end date (up to +/- 6 months, coloured lines) around each SPOT visit (X axis at 0) of each of the 22 Amnesty POIs (cell), in growing order of latitude (figured at the top of each cell).

It shows that we could potentially try to pick POIs and SPOT visit times that maximise the number of Sentinel-2 imagery available within a fixed length time window, so as to have the richest training set. We do indeed observe some variation in that regard: some of these lines go much higher than others.

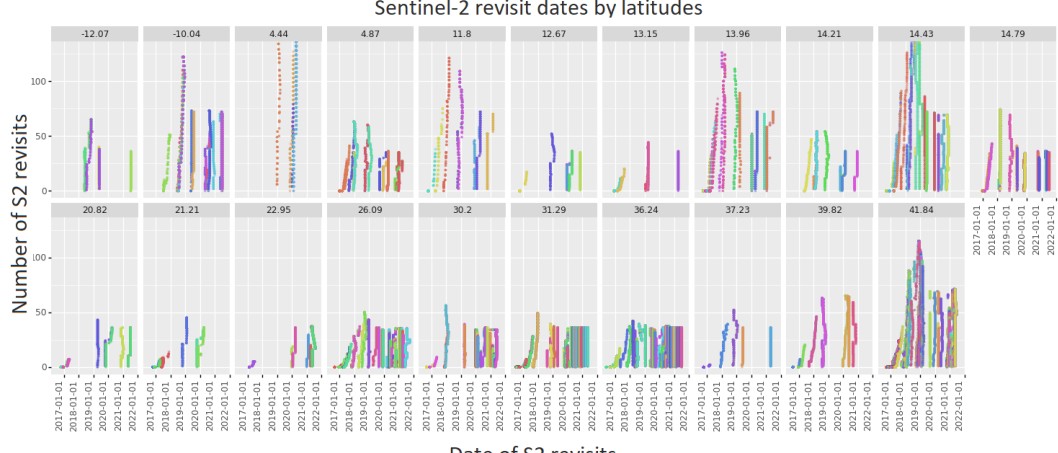

Figure 4: Number of Sentinel-2 visits for each SPOT visits on Amnesty International's AOIs. Each graph corresponds to an AOI and is labeled by its latitude at the top, to visualise the effect of location on revisits availability.

However, this would induce an implicit bias of a nature hard to interpret. We also observe that within a POI, the discrepancy between the number of S2 revisits, while clearly present, is reasonable, with multiple SPOT revisits offering similar S2 availability.

Finally, biasing per Sentinel-2 imagery would be akin to biasing per Sentinel-2 cloud coverage: this would not be a fair representation of real-world use cases, and we would therefore be training our models for the wrong problem.

We therefore took the decision yet again to solve a harder problem than an optimally-curated dataset would make for, so as to be the closest to reality. To that effect, within a POI, we pick uniformly at random the SPOT visit to use as a reference.

Of course, one bias remains: we will not have imagery of POIs that have never been tasked by SPOT customers. While unfortunate, there is no way around it, short of using another high resolution product. We do have hope in two mitigating factors:

- SPOT swath covers more than just the single POI, so we cover areas that are possibly more diverse than just the one precise point of interest to the SPOT customer.
- SPOT tasking means the POI exhibits features of activity interesting to at least the SPOT customer. SPOT customers might not have entirely the same interests as the users of our open-source package, but it is not unreasonable to assume that the features will be transferable. Therefore, this implicit sampling is actually a positive way to ensure interesting features.

## 4 Putting it to Use: Baselines, Benchmarks, and Source Code

We believe this WorldStrat dataset can enable a broad range of applications. While we benchmark here multi-frame superresolution, closest to the authors' own expertise, this is by no means restrictive. More applications can be devised, either with extra labelling, or using self-supervised representations on low and high resolutions, e.g. extending Tile2Vec Jean et al. (2019), or even learning transfer tasks from one resolution to the other.

### 4.1 Super-Resolution Benchmark

We illustrate the use of this dataset on the task of superresolution. While there has been considerable recent progress in multi-frame super-resolution (see Salvetti et al. (2020); Valsesia and Magli (2022); Bhat et al. (2021); Molini et al. (2020)), since this is not meant as an exhaustive benchmark article, we

only focus on three architectures: 1) the single-image super-resolution architecture SRCNN (Dong et al., 2015); 2) our multi-frame extension of SRCNN (super-resolution convolutional neural network) by collating revisits as channels; and 3) a multi-spectral modification of the original HighResNet (Deudon et al., 2020) to handle multiple bands similarly to (Razzak et al., 2021).

We accelerated HighResNet by replacing the learned ShiftNet by a simple cached alignment search. The two core architectures for (multi-frame) SRCNN and HighResNet are visualised in Figure 5.

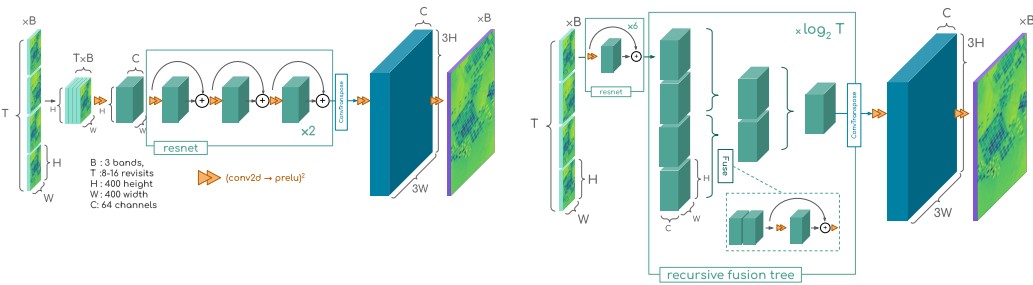

Figure 5: Multi-frame super-resolution architectures. Left: Multi-Frame SRCNN. Right: HighResNet. Single-frame SRCCN is achieved by using Multi-Frame SRCNN with a single revisit.

All implementations are our own, and the code as well as the trained models are released as part of our eo-learn plugin, described in the following section. Figure 6 shows the results.

We use the metrics established by Märtens et al. (2019). Those consist of a weighted combination of mean average error (MAE), mean squared error (MSE), peak signal-to-noise ratio (PSNR) and structural similarity index (SSIM).

For each of the three architectures, we trained three models using three varying fixed random seeds, to account for results variability within an architecture. The same three seeds were used for every architecture, and a single fixed random seed was used for all data loading and processing operations, to ensure comparability between the architectures.

All training runs were done over the entire dataset, using 12-bit radiometry and 8 low-resolution revisits, using the published train, validation and test splits. Class stratification is ensured within the train, validation and test splits. We used the Adam optimizer with a Cosine Annealing Warm Restart scheduler (Loshchilov and Hutter, 2017). We also provide Stochastic Gradient Descent (SGD) as an alternative optimizer.

The benchmarks are reproducible using the provided open-source toolbox, and we provide all the hyperparameters in the Supplementary Material.

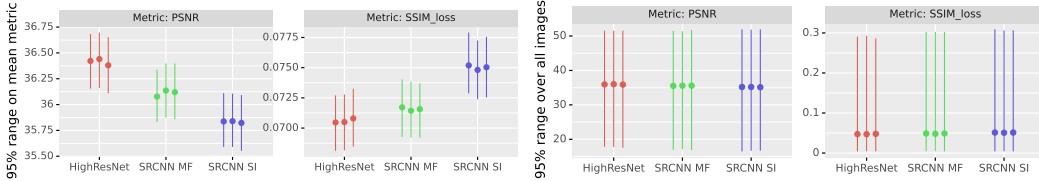

Figure 6: Comparing three architectures and two metrics over the validation set for three independent training runs. PSNR (Peak Signal-to-noise Ratio): higher is better. SSIM (Structural Similarity Index): lower is better. Left: comparing the 95% Confidence Intervals (CI) of the estimation of the mean of the metric. Right: comparing the 95% range of the metric over the whole validation set. Note how, while the means seem significantly different, the variability across the distribution absolutely dwarfs any impact of the algorithms, pointing to the need for moving away from means-based benchmark, and to the need for better algorithms.

Rather than providing just the mean of each metric over the validation set (possibly with 95% confidence intervals), we decide here to follow a harder but, we believe, much more rigorous

reporting: we report the *95% range of the metric over the whole validation set*, i.e. the 2.5-percentile and the 97.5-percentiles over all data points in the validation sample. This provides for a much more rigorous comparison of algorithms, by showing the full context of the metric over the whole distribution of inputs. The reader can thus better see whether a few extra points on a metric make a strong difference or are just minor compared to within-distribution variability.

As a side note, ablation studies (not shown here) have confirmed that the three above architectures enjoy a clear improvement when going from 4 revisits to 8, but only minor diminishing returns when increasing to 16. Another ablation study on the size of the training set shows there is still room for improvement were extra budget WorldStrat to be extended.

### 4.2 Toolbox and eo-learn Plugin

In addition to the dataset, we open-source a Python package designed for ease of use. It integrates with the widely eo-learn Python package as one in eo-learn pipeline, and provides abundant tutorial notebooks. We made this dataset to be reproducible and extensible, and thus include notebooks covering data collection, training, and inference of all baselines, with standardized interfaces in Pytorch Lightning. We provide the ability to sample new training data as needed. Particular care was paid to High-Efficiency Training, to make this world accessible with modest computing budget: our variant of HighResNet **trains in 30 minutes on a single V100 GPU**, thanks to our fast caching mechanism for 95% average GPU usage.

## 5  Discussion

**Potential social impact:** A whole coverage of the potential social impact of computer vision and remote sensing could cover a whole book. However, in our specific case, the only sensitive data are the UNHCR locations, but these have already been released by UNHCR themselves.

**Known limitations and future work:** Although we want a broad range of applications to be possible with our trained models, we know that the frequency of revisits constrains us to structures that change at a slower pace than Sentinel-2 revisits – one revisit every five days. This puts more emphasis on permanent structures, such as buildings or a form of land occupation, and rules out higher temporal variability use cases, such as immediate state of crops.

We did not enrich the dataset with rivers/harbours and liminal coastal space. This could be done by finding maps of water courses, and selecting locations at these intersections.

Stratification could be endlessly refined. For example we considered but discarded stratifying by use of the Local Climate Zones (LCZ) database of the World Urban Database (Stewart and Oke, 2012). LCZ tries to identify the type of buildings on a local scale, and is available for Europe, The Americas, and contributed zones (Demuzere et al., 2021). However, the coverage of the contributed zones is not as wide as the GHSL.

The UNHCR Persons of Concerns dataset is not guaranteed to be extremely precise: its POIs are more likely "in the neighbourhood" of the actual settlements. We mitigate this by covering a 2.5 km² area, and by accounting for the fact that temporary settlements are often similar to the surrounding settlements, which are therefore representative.

## Acknowledgments and Disclosure of Funding

Project empowered by the ESA Phi Lab (https://philab.phi.esa.int) as part of the ESA-funded QueryPlanet project 4000124792/18/I-BG CCN3.

We are very grateful to Pierre-Philippe Matthieu (ESA Phi-Lab) and Nicolas Longepe (ESA Phi-Lab) for their support and belief in this project. To Grega Milcinski (Sinergise) and SentinelHub for taking us on board QueryPlanet. To Jamon Van Den Hoek (Oregon State University) for his expertise on GHSL and providing the UNHCR POIs dataset. To Micah Farfour (Amnesty International) for POIs of humanitarian interest. To Moritz Besser (dida Datenschmiede GmbH) for their ASMSpotter data. And Peggy Fischer, Bryan Keary, and Montserrat Del Riego (ESA TPM) for their support in making the imagery available publicly.

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
