# OpenReview forum: "Open High-Resolution Satellite Imagery: The WorldStrat Dataset – With Application to Super-Resolution"
_NeurIPS.cc/2022/Track/Datasets_and_Benchmarks — NeurIPS 2022 Datasets and Benchmarks _

### Official Review · Reviewer_33Vk · 2022-07-16
**assessment of paper as potential NeurIPS dataset contribution**

**Rating:** 5
**Confidence:** 5

**Strengths:**

- standardized code (PyTorch Lightning + integration to existing EO-Learn Python library) allows to verify and flexibly extend in future the dataset
- well-designed geospatial sampling strategy to maintain diversity of satellite imagery

**Weaknesses:**

As noted by the authors in Sect. 5, the dataset is limited to little cover features over water bodies, oceans, in particular. Consequently, it limits applications for vessel detection, fish farming, oil spills, etc.
Further, as it stands, without additional layers for e.g. semantic segmentation, the dataset is most valuable for super-resolution or unsupervised learning approaches. Regarding the term *high-resolution*, I suggest the authors expliticly note that Sentinel-2 and SPOT do not significantly differ without pansharpening techniques, i.e. 10m vs. 6m.
Moreover, I recommend to cut down the abstract and the main text to the point, reducing loose phrases in favor for more crisp statements, e.g. compare the lengths of the introduction, Sect. 1.1 vs. the size of the conclusions, Sect. 5.

**Additional Feedback:**

I highly value the efforts of the authors to make openly available a curated remote sensing data set as presented by Cornebise et al. However, the work needs adjustments according to the above before publication. The manuscript's scope for NeurIPS is rather restrictive to super-resolution of satellite imagery. Applications beyond require additional efforts such as geospatial data fusion in order to assemble the required labels. In particular, such efforts pose a barrier to researchers not skilled in processing earth observation data - even when convenient libraries such as EO-learn do exist.

**Clarity:**

kindly:
- In the interest of a neutral research approach, please remove emotional phrases such as:
    * _Computer vision and satellite imagery seem to be a match made in heaven._, What does define the term *heaven*?
    * _from this admittedly elaborate figure_, What criterion does define *elaborate*?
    * avoid the use of exclamation marks
- increase of annotation size in Fig. 3 improves reading
- please simplify & summarize Fig. 4
- properly introducing acronyms such as AOI (area of interest), POI (point of interest), etc. for easier access of researchers outside the scope of remote sensing

**Correctness:**

please:
- drop (illegal?) Meteosat signal streaming story
- reference or drop vague statements such as *possibly the most widely used satellite imagery datasets*
- minor: please adjust typo *percent o will*

**Documentation:**

Please add web-links to code (GitHub) and data (Zenodo) to the paper. I was unable to spot those in the supplemental material, too.

**Ethics:**

The dataset contains imagery of spatio-temporal points-of-interest reported for illegal mining and related to areas-of-interest for Amnesty International. Please assess risks related to legal consequences for humans on the ground exposed to supressive governments, etc.

**Relation To Prior Work:**

I suggest:
- cite review of computer vision in remote sensing for multi-spectral imagery, e.g. from IEEE Geoscience & Remote Sensing journal
- list available geospatial data platforms beyond Google Earth
- a more comprehensive list of super-resolution datasets in general
- inclusion of recent developments in deep learning on super-resolution techniques is recommended;
  in particular this year's CVPR did present relevant works wrt. pansharpening

**Summary And Contributions:**

Cornebise, Orsolic, and Kalaitzis in "Free High-Resolution Imagery: The WorldStrat Dataset" present a medium-resolution (approx. 6m per pixel), multi-spectral (RGB, NIR) SPOT-satellite imagery dataset including a high-resolution (approx. 1.5m per pixel) equivalent through pansharpening, and a spatially-matched, temporally SPOT-centered timeseries of 16 twelve-band, 10m (RGB, NIR) to 60m (others) per pixel multi-spectral images as captured by the Sentinel-2 satellites. Sampling of data is with focus over our planet's land  based on three generic use case scenarios for Earth observation applications. The dataset will ship with code to pull the data online, and to perform an image super-resolution benckmark utilizing 3 corresponding deep neural network architectures.

---

> ### Author Response · Authors · 2022-08-23
> **Answer to reviewer 33Vk (5/10, 5/5) (part 1/2)**
>
>
> ### Request to the AC
>
> We cannot help but observe that this review, although the _only rejection_ out of six reviews, is also the one with the highest self-confidence score, being "absolutely certain". It is also the reviewer with the more derogatory tone, surprising and out-of-place suggestions of illegality of our past work, lack of familiarity with the Datasheet for Dataset framework (required by the Track itself), and expressing clear judgements of taste.
>
> In this light, and while we will take care to answer each of the points raised by the reviewer, **we respectfully request that the AC discards this review**.
>
> ### Answers to the reviewer
>
> Thank you for your time reviewing this article. We address here the specific points you raised, and invite you to also read the [common answer part 1](https://openreview.net/forum?id=DEigo9L8xZA&noteId=CIl-pnhE7Gn) and its [part 2](https://openreview.net/forum?id=DEigo9L8xZA&noteId=xVA9llVJqkX).
>
>
> > Regarding the term high-resolution, I suggest the authors expliticly note that Sentinel-2 and SPOT do not significantly differ without pansharpening techniques, i.e. 10m vs. 6m.
>
> True. We argue that the presence of the panchromatic channel (PAN), however, is a very important difference. It allows for a much richer spectral deconvolution process, and we believe the joint distribution accross channels will bring value to algorithms. We do know that PAN is not suited for applications requiring a very precise spectral resolution, like scientific measurements. However, when it comes to Earth observation for socio-geographical applications, we have seen countless users (including our partners) relying heavily on the panchromatic band for the added details it provides.
>
> We also note that, in our super-resolution application, super-resolving to full PAN-sharpened resolution actually proved much difficult, and we targeted instead a de-facto 3x zoom from 10m to 3.3m.
>
> As such, in the absence of the dream data of sub-meter imagery on all bands, we are comfortable providing PAN and believe in the value of this dataset.
>
> > Moreover, I recommend to cut down the abstract and the main text to the point, reducing loose phrases in favor for more crisp statements, e.g. compare the lengths of the introduction, Sect. 1.1 vs. the size of the conclusions, Sect. 5.
>
>  We appreciate that other reviewers have lauded the amount of motivation provided in this article. We do believe that scholars, a fortiori in a field now so broad as machine learning, can appreciate story-telling and a style possibly more conversational.
>
> > drop (illegal?) Meteosat signal streaming story
>
> We are very surprised that the reviewer questions the legality of this past work. I believe it is the first time in 17 years of research that I see such a comment in a review, to my own articles or to articles that I have reviewed.
>
> For the interested reader, and if memory serves 17 years after the fact, Meteosat in 2005 required a license for acquiring the (encrypted) digital signal, but made the analog signal widely available.
>
>
> > reference or drop vague statements such as possibly the most widely used satellite imagery datasets
>
> It is clear from this and other comments that the reviewer does not appreciate the writing style. I hope for the reviewer that they never have to read the [YOLOv3 article](https://arxiv.org/abs/1804.02767), cited 14,722 times as of writing, which goes much further than we do.
>
>
> > minor: please adjust typo percent o will
>
> Thank you for flagging it.
>
> > In the interest of a neutral research approach, please remove emotional phrases such as: Computer vision and satellite imagery seem to be a match made in heaven., What does define the term heaven?
>
> We have used "Made in heaven" is its colloquial sense [defined by Merriam-Webster defines](https://www.merriam-webster.com/dictionary/made%20in%20heaven) as "very good and successful". There are precedents for the use of this expression in a scientific scholarly setting, for example [Sandor and Kosakov (2013)](https://www.ncbi.nlm.nih.gov/pmc/articles/PMC3942495/).
>
>
> > From this admittedly elaborate figure, What criterion does define elaborate?
>
> Elaborate is [defined by Merriam-Webster](https://www.merriam-webster.com/dictionary/elaborate) as "marked by complexity, fullness of detail, or ornateness". We believe that Figure 4 is complex and full of details, and thus not easy to read at first glance, since it is an atypical visualisation. We hereby wanted  to provide the reader with reassurance that they are not alone in needing to read it a few times to make sense of it.
>
> We have now replaced that mark of empathy by what we believe is a clearer explanation of that figure.
>
> **(continued in next post due to character limit)**

---

> > ### Author Response · Authors · 2022-08-23
> > **Answer to reviewer 33Vk (5/10, 5/5) (part 2/2)**
> >
> >
> > **(continued from previous post due to character limit)**
> >
> > > Increase of annotation size in Fig. 3 improves reading.
> >
> > Thank you for pointing this. This will be done for the camera-ready version.
> >
> > > Please simplify & summarize Fig. 4.
> >
> > We have added a summary and explanations in the main text.
> >
> > > Properly introducing acronyms such as AOI (area of interest), POI (point of interest), etc. for easier access of researchers outside the scope of remote sensing.
> >
> > This has been addressed and all acronyms are now properly introduced.
> >
> > > cite review of computer vision in remote sensing for multi-spectral imagery, e.g. from IEEE Geoscience & Remote Sensing journal
> >
> > We will be glad to cite any such review the reviewer deems the most suitable, rather than a reference to a journal.
> >
> > > list available geospatial data platforms beyond Google Earth
> >
> > The reviewer seems to have overlooked that we make heavy use and mention of SentinelHub. Had the reviewer also checked the code, or the notebooks, they would have observed that the entire compatibility is with SentinelHub, not with Google Earth.
> >
> > > a more comprehensive list of super-resolution datasets in general
> > > inclusion of recent developments in deep learning on super-resolution techniques is recommended; in particular this year's CVPR did present relevant works wrt. pansharpening
> >
> > We invite the reviewer to compare the submission deadline for this track and the dates of CVPR, and then explain how we could have cited articles from what was then a *future conference*.
> >
> > > Please add web-links to code (GitHub) and data (Zenodo) to the paper. I was unable to spot those in the supplemental material, too.
> >
> > The links are (now) present in the abstract of the article. Depending on when the reviewer first accessed the article, and as noted in the initial communication with reviewers, there was a short window of time when the data itself was not available on Zenodo, but rather on Google Drive for reviewing purposes initially, due to the negotiations between ESA and Zenodo taking place.
> >
> > > The dataset contains imagery of spatio-temporal points-of-interest reported for illegal mining and related to areas-of-interest for Amnesty International. Please assess risks related to legal consequences for humans on the ground exposed to supressive governments, etc.
> >
> > We are glad that the reviewer has these concerns in mind, like we do.
> >
> > We respectfully suggest that the reviewer familiarizes themselve with the Datasheet for Datasets framework, which is part and parcel of the requirements for this track. They would then know to check there for our detailed discussion of these ethical concerns.

---

> > > ### Comment · Reviewer_33Vk · 2022-09-02
> > > **I am ok with publication if the AC is, but stay with my rating**
> > >
> > > Dear authors:
> > >
> > > I am sorry if any of my statements hit a nerve - that's at least what I read off your emotional comments. I definitely appreciate your hard work.
> > > My statements is mostly suggestions and questions. Note that I rate marginally below acceptance, not a clear reject.
> > >
> > > Thank you for taking the effort to address a subset of my comments. Given the increasing amount of papers published daily in the domain of AI and geospatial, I did suggest to cut down the paper's length to the bare minimum for our colleagues to rapidly digest. Supplementary material provides an opportunity for in-depth and additional story telling.
> > >
> > > Provided the camera-ready version of your paper is going to underline
> > >
> > > > Regarding the term high-resolution, I suggest the authors expliticly note that Sentinel-2 and SPOT do not significantly differ without pansharpening techniques, i.e. 10m vs. 6m.
> > >
> > > to avoid any misleading statements about the data resolution plus some discussion on limits of pansharpened images, I am ok with acceptance of the paper given the other reviewers indicate their ok.
> > >
> > > Thank you again for pushing remote sensing analytics.

---

### Official Review · Reviewer_ao84 · 2022-07-22
**A solid super-resolution dataset, not only fitting the urging needs of the remote sensing community but also being attractive to a broader audience**

**Rating:** 7
**Confidence:** 5

**Strengths:**

### Significance:
The submission provides a dataset of high resolution satellite data, which is attractive for remote sensing researchers (commonly limited by access constraints to low-to-medium resolution Earth observations) as well as the computer vision community (oftentimes focusing on more familiar very-high resolution UAV RGB data). High resolution satellite imagery is beneficial yet resource-consuming to obtain. Hence, the work’s significance is twofold: in making such images accessible over diverse and interesting areas, as well as in providing a super-resolution benchmark dataset on said observations.

### Relevance:
* Super-resolution is a competitive and active field in satellite imagery, underlining the community’s needs for observations of a finer spatial resolution than is commonly available.
* The authors provide a global sampling following an in-depth geospatial stratification of areas, in order to accomplish a representative, diverse and interesting distribution of sites. The geospatial sampling is motivated in great detail and also regards typically under-represented land cover classes. The wide and heterogeneous coverage ensures the gathered data’s relevance for researchers focused on particular regions as well as those interested in whole-planet observations.
* Machine Learning for social good is becoming a theme of increasing interest, and I am positive that the community will find value in the dataset’s curated high-resolution samples over critical sites, as identified by e.g. UNHCR and Amnesty.

### Accessibility and Accountability:
The authors do an outstandingly great job at making their contribution accessible. The dataset is hosted on zenodo, which I consider an appropriate platform. The submission comes with an associated python package, facilitating the distribution and access of the dataset. The provided documentation  follows the [“Datasheets for Datasets”](https://arxiv.org/abs/1803.09010) template, is in great detail and goes beyond the standards of comparable submissions. This is welcome, as multi-spectral remote sensing may still be novel to (parts of) the computer vision community and hence benefit of further explanations as provided therein.

### Ethical and social implications:
* The authors clarify that the utilized land use and land cover products, albeit containing potentially sensitive information (e.g. wr.t. refugee settlements), have readily been released by the corresponding parties (e.g. UNHCR). Furthermore, the authors consulted an independent to assess providing high-resolution imagery of potentially sensitive land cover may not entail any negative social implications.

**Weaknesses:**

### Significance:
* The core idea and motif of the submitted dataset is fairly well-established. Submissions like [Märtens et al., 2019] and the very recent work of Michel et al (2022) are closely related. While the improvements on prior work are thus of a rather quantitative nature, the submitted dataset convinces in its quality of execution (e.g. careful stratification of areas etc).
* The experimental evaluation in the submission is kept fairly brief. With 3 models evaluated on the proposed data, the benchmarking is relatively compact, compared to other dataset submissions. The authors reference related multi-frame methods in section 4.1 and there’s additional single-frame methods that are of relevance [i, ii]. Super-resolution is a competitive and active field, and any extensive and readily available benchmarking may facilitate the prospects of the community adapting a novel dataset. Hence I would appreciate the author’s comments on whether they plan to extend the current evaluations (by e.g. benchmarking further models and conducting ablation studies on data) following an eventual acceptance or in the scope of future works.

### Relevance:
* The submission focuses on super-resolution in satellite imagery, which may be a more specific and rather novel domain for the broader super-resolution in computer vision community. But the submission does a good job of making such data more approachable and thereby contributing to the growing relevance of Earth observation to a broader audience.
* Earth observation imagery often has spatio-spectral characteristics specific to a particular instrument. That is, the provided SPOT 6/7 high-resolution data may be significantly different from e.g. the 0.5m resolution Maxar images provided in the SpaceNet challenge datasets [Van Etten et al., 2018] or the 5m resolution VENμS satellites in Michel et al. [2022]. Likewise, the spatio-spectral properties of Sentinel-2 differ from e.g. the commonly utilized Landsat missions’ sensors. In that sense, any upsampling facilitated by the submission will necessarily be sensor-specific and does not straightforwardly apply to other instruments.

###  Accessibility and Accountability:
While the authors do a good job making the dataset easily accessible and thereby facilitating its future adaptation, I do have some concerns about the data volume and whether the entirety of the included data is properly motivated. For instance, it is not apparent what the added benefits of the total 8+8 medium resolution observations are over e.g. a smaller set of associated observations. At best, this surplus of data may have benefited from a proper motivation backed by ablation experiments. At worst, this makes the dataset unnecessarily bloated and voluminous in file size without significant benefits. To avoid accidentally excluding researchers from working on the curated dataset due to potential computational constraints, I would suggest the authors address this issue by making it an option to solely download and extract single-frame data. The authors have readily provided a ‘Core’ version of their dataset on kaggle, containing only level 2A images and 8 rather than 16 medium-resolution samples. Further ablation experiments to motivate the full dataset and an option to access only single-frame data for works the likes of [i, ii] would be very much welcome.

### Ethical and social implications:
As super-resolution is an ill-posed problem, reconstructing the signal could result in biased outcomes as [previously](https://twitter.com/Chicken3gg/status/1274314622447820801) reported in the community. I do consider satellite surveying being a potential risk application in these regards. While the authors did not specifically discuss the raised concern, I do however think that their data curation approach (at least to a sufficient extent) should resolve this issue: a) the dataset is sampled globally to obtain very heterogeneous areas b) including under-represented areas (shelters etc). And finally, c) multi-temporal medium-resolution observations may act as regularizers to learning the reverse mapping. I am positive the authors may provide further arguments to address the raised concern.


[i] Lanaras, C., Bioucas-Dias, J., Galliani, S., Baltsavias, E., & Schindler, K. (2018). Super-resolution of Sentinel-2 images: Learning a globally applicable deep neural network. ISPRS Journal of Photogrammetry and Remote Sensing, 146, 305-319.

[ii] Pouliot, D., Latifovic, R., Pasher, J., & Duffe, J. (2018). Landsat super-resolution enhancement using convolution neural networks and Sentinel-2 for training. Remote Sensing, 10(3), 394.

**Additional Feedback:**

The authors make the design choice to include medium-resolution satellite data featuring clouds. While arguably better reflecting real use cases, this somewhat mixes a pure super-resolution task with a denoising and image reconstruction task. I find this approach principally justified, but have some follow-up questions:
* What are the statistics on cloud coverage like, what are the mean and variance of cloud coverage? Do you observe considerable differences across some sampled regions or land cover types?
* Are you planning to provide pixel-wise cloud mask annotations for mask-guided reconstruction and super-resolution approaches?
* It would be great to provide code for evaluating on a distinct subset of the dataset exclusively containing cloud-free samples, in order to allow for focusing on the super-resolution task in isolation. Controlling for sample numbers while involving cloudy observations in a separate condition would make for an interesting comparison. This may provide insights about the difficulty added by the cloud-covered observations.

**Clarity:**

The paper is well written. While always remaining clear and understandable, the writing occasionally follows a rather informal style and appears somewhat quirky at times. Its okay as is, but I nonetheless suggest the authors make some minor edits to e.g. the Introduction section’s writing style.

**Correctness:**

The claims are correct and sustained. However, I would like to add the following minor remark: Copernicus Sentinel 2, according to [ESA’s user guide](https://sentinels.copernicus.eu/web/sentinel/user-guides/sentinel-2-msi/overview) does actually come with 13 rather than 12 bands. Hence, the curated dataset does technically not contain the fulls spectrum of Sentinel 2 imagery, other than stated in the submission. As far as I may infer, the missing band is 10 (Cirrus), which may have been dropped by to SentinelHub’s level 2A preprocessing pipeline. However, I think this does not overly impact the submission, as the the missing band has a 60 m resolution and its primary purpose of atmospheric screening is mostly irrelevant for the super-resolution task.  Nonetheless, I suggest the authors either add the 13th band or clarify this in the main text. As I do consider this work of potential interest for the wider computer vision community, which may not be familiar with remote sensing instruments, it would be very welcome to avoid potential future confusions about the 12/13-bands matter.

**Documentation:**

Yes to all. The dataset is very well documented and I do consider this a strength of the submission. The provided documentation follows the [“Datasheets for Datasets”](https://arxiv.org/abs/1803.09010) template, is in great detail and goes beyond the standards of comparable submissions. This is welcome, as multi-spectral remote sensing may still be novel to (parts of) the computer vision community and hence benefit of further explanations as provided therein.

**Ethics:**

Please see “Ethical and social implications” under section “Weaknesses”.

**Relation To Prior Work:**

Relevant publications are cited and discussed in sections 1 and 4.1. The closest prior work is probably a dataset on single-image super-resolution by Michel et al. (2022), which was published very recently and close to the deadline however. While the given submission is similar in its spirit, it considerably extends on the scope, volume and diversity of preceding datasets.

**Summary And Contributions:**

The submission provides a **novel dataset for super-resolution in optical satellite images**. The dataset consists of repeated measures of medium resolution Sentinel 2 and a high resolution SPOT 6/7 reference satellite observation per sample. Key contributions are highlighted in **bold** font.

This is a solid contribution and I recommend for acceptance. The work may advance super-resolution approaches on satellite imagery---which is a popular topic, underlining the community’s needs for higher-resolution data. The authors do a great job of documenting their data and making it easily accessible, which raises the expectations that the submission may become of interest to the broader computer vision community. Overall, the idea and the motif underlying this submission may be well-established and there exists related prior work, but the execution of curating and documenting the dataset at hand is outstandingly well done.

---

> ### Author Response · Authors · 2022-08-23
> **Answer to reviewer ao84 (7/10, 5/5) (part 1/3)**
>
>
>
> Thank you very much for your review. We address here the specific points you raised, and invite you to also read the [common answer part 1](https://openreview.net/forum?id=DEigo9L8xZA&noteId=CIl-pnhE7Gn) and its [part 2](https://openreview.net/forum?id=DEigo9L8xZA&noteId=xVA9llVJqkX).
>
>
> > I would appreciate the author’s comments on whether they plan to extend the current evaluations (by e.g. benchmarking further models and conducting ablation studies on data) following an eventual acceptance or in the scope of future works.
>
> We hope the common answer addressed this specific concern! We are glad that this dataset-focused article and the associated "bonus" benchmark seemed rigorous enough to warrant further extension of the benchmark. We did perform ablation studies several times during the project, and now briefly discuss them in the article.
>
> As to benchmarking future models, similar to, e.g., the ImageNet dataset for computer vision, we do hope that WorldStrat becomes a standard in machine learning for Earth observation and will be used by new works to report their performance. This is why we provide abundant code for reusability and reproducibility, and the most permissive licenses possible.
>
> We also welcome any contribution via the Github repository, if users want to contribute back their own benchmarks or implementations!
>
> >  I do have some concerns about the data volume and whether the entirety of the included data is properly motivated. For instance, it is not apparent what the added benefits of the total 8+8 medium resolution observations are over e.g. a smaller set of associated observations.
> >
> >  At best, this surplus of data may have benefited from a proper motivation backed by ablation experiments. At worst, this makes the dataset unnecessarily bloated and voluminous in file size without significant benefits. To avoid accidentally excluding researchers from working on the curated dataset due to potential computational constraints, I would suggest the authors address this issue by making it an option to solely download and extract single-frame data.
>
> This is a very important point, and we are delighted that you raise it. Too many of our colleagues with restricted bandwidth cannot afford to download or operate on large datasets. This has been at the forefront of our mind throughout this work, as can be seen for example in our focus of algorithms such as HighResNet that can train on relatively modest computing budgets (a few hours on a single GPUs).
>
> We also debated as to whether 16 revisits were too heavy. In the initial stages of the project, we downloaded 32 medium resolution observations. After running ablation studies on the HighResNet model, we found that 8 revisits were the best trade-off between compute requirements and gained accuracy.  Using 16 revisits does yield marginally better results even on HighResNet, but coupled with a 50% memory requirement increase. Since we optimised our benchmarks for both compute efficiency as well as accuracy, we decided to use 8 revisits for the benchmark.
>
> We do release the 16 revisits as we do not want to presume of other algorithms' behaviours. But we did not include 32, to avoid being overkill.
> Considering the dataset was designed for a broad spectrum of uses, and that our benchmarks most certainly won’t be the largest or most sophisticated super-resolution architectures going forward, we included the additional 8 revisits to make further research and development as easy as possible. Should a need arise for even more revisits, the open-source toolbox allows for simple extension of the dataset, and we felt that more than 16 revisits would be unwarranted, per the ablation studies results.
>
> That being said, we agree with your assessment that it would be much better to allow the download of single-frame data. This is something we’ve started to explore right after the release of the dataset.  It was pointed out to us by the community that [Cloud optimised GeoTIFFs (COGS)](https://github.com/cogeotiff/cog-spec/blob/master/spec.md) might be a good solution. They provide a way to download and extract even parts of a single frame, using the [Range Requests feature of HTTP/1.1](https://www.rfc-editor.org/rfc/rfc7233).  The downside is that the data cannot be compressed, which requires us to negotiate with Zenodo to provide us with even more free hosting space. [Zenodo has luckily implemented the Range Requests feature as of November 2021, so this is a possibility.](https://github.com/zenodo/zenodo/issues/1599)
>
> Finally, these accessibility concerns are also why we uploaded the dataset not just in compressed form to Zenodo, but also in uncompressed form to Kaggle, where any user can spin up a cloud-hosted notebook without having to download anything or own any local compute beside their web browser.
>
> **(continued in two next posts due to character limit)**

---

> > ### Author Response · Authors · 2022-08-23
> > **Answer to reviewer ao84 (7/10, 5/5) (part 2/3)**
> >
> >
> > **(continued from previous post due to character limit)**
> >
> > > As super-resolution is an ill-posed problem, reconstructing the signal could result in biased outcomes as previously reported in the community. I do consider satellite surveying being a potential risk application in these regards.
> > >
> > > While the authors did not specifically discuss the raised concern, I do however think that their data curation approach (at least to a sufficient extent) should resolve this issue: a) the dataset is sampled globally to obtain very heterogeneous areas b) including under-represented areas (shelters etc). And finally, c) multi-temporal medium-resolution observations may act as regularizers to learning the reverse mapping.
> > >
> > > **I am positive the authors may provide further arguments to address the raised concern.**
> >
> > This concern is absolutely crucial! We could discuss for pages and pages about the risk biased outcomes, risks of "hallucinations", and how better metrics are needed to prevent those. As a matter of fact, the third author is leading a consortium on that very topic of safe super-resolution, which obtained a multi-year funding from the European Space Agency starting this summer, so expect to see a lot of progress in the area.
> >
> > Now, within the more limited context of the WorldStrat article, we did want to focus on the broad representativity, which as you perfectly argue, should help some in this regard. We did not want to over emphasize super-resolution solely in the article, as we believe the dataset can be used for other research endeavours too (as now specified in the beginning Section 4). These concerns are so important that they deserve an in-depth treatment in their own article.
> >
> >
> > > The claims are correct and sustained. However, I would like to add the following minor remark: Copernicus Sentinel 2, according to ESA’s user guide does actually come with 13 rather than 12 bands.
> > >
> > > Hence, the curated dataset does technically not contain the fulls spectrum of Sentinel 2 imagery, other than stated in the submission. As far as I may infer, the missing band is 10 (Cirrus), which may have been dropped by to SentinelHub’s level 2A preprocessing pipeline.
> > >
> > > However, I think this does not overly impact the submission, as the the missing band has a 60 m resolution and its primary purpose of atmospheric screening is mostly irrelevant for the super-resolution task.
> > >
> > > Nonetheless, I suggest the authors either add the 13th band or clarify this in the main text. As I do consider this work of potential interest for the wider computer vision community, which may not be familiar with remote sensing instruments, it would be very welcome to avoid potential future confusions about the 12/13-bands matter.
> >
> > We are delighted to see a reviewer who knows and cares for the differences between bands.
> >
> > Sentinel-2 imagery comes in two product types:
> > - **Level-1C (L1C)** which provides top-of-atmosphere (TOA) reflectance images, and
> > - **Level-2A (L2A)** which provides bottom-of-atmosphere (BOA) reflectance images derived from the associated Level-1C products by the European Space Agency.
> >
> > Depending on the product type, Copernicus Sentinel-2 comes with 13 (L1C) or 12 bands (L2A).
> >
> > You correctly inferred that L2A omits the 10th (cirrus) band, [as noted in the Sentinel-2 Processing guide](https://sentinels.copernicus.eu/web/sentinel/user-guides/sentinel-2-msi/processing-levels/level-2).
> >
> > **The dataset contains both product types and all of their bands (13 for L1C and 12 for L2A)**, but since we mainly used L2A, the main article stated that we include all 12 bands, as you’ve correctly pointed out.
> >
> > We have now addressed this in Section 3.2. by stating that the low-resolution imagery comes in these two product types, containing a different number of bands, and that we provide both product types fully. Thank you for pointing this out.
> >
> >
> > > The paper is well written. While always remaining clear and understandable, the writing occasionally follows a rather informal style and appears somewhat quirky at times. Its okay as is, but I nonetheless suggest the authors make some minor edits to e.g. the Introduction section’s writing style.
> >
> > Thank you very much for the compliments. As discussed in the common answer, we do acknowledge that the writing is in places somewhat unconventional, partly as a daring (and possibly controversial) choice to broaden the accessibility via a less dry writing style than typical. Yet we do not want the form to distract from the content. Would you be please be able to point to the precise wordings that you find the most jarring, so we can address those?
> >
> >
> > **(continued in next and final post due to character limit)**

---

> > > ### Author Response · Authors · 2022-08-23
> > > **Answer to reviewer ao84 (7/10, 5/5) (part 3/3)**
> > >
> > >
> > > **(continued from previous post due to character limit)**
> > >
> > > > What are the statistics on cloud coverage like, what are the mean and variance of cloud coverage? Do you observe considerable differences across some sampled regions or land cover types?
> > >
> > > Thank you for this question!  We have added our answer below in an extra Appendix, as indeed these are very fair questions!
> > >
> > > The mean of the cloud coverage is 7.98, and the variance is 202.28.
> > > The quantiles are:
> > > - 0.025: 0.00
> > > - 0.25: 0.00
> > > - 0.5: 0.66
> > > - 0.75: 10.05
> > > - 0.975: 49.95
> > >
> > > It's important to note that this cloud cover percentage, as mentioned in the article and datasheet, is calculated on the entire product size of the provider, which varies in size but is much larger than the 2.5km² we target. This means that even an image with a large cloud cover percentage can be cloud free, and in extreme cases (though unlikely), vice-versa.
> > >
> > > Also, as you might imagine, there indeed are considerable difference across sampled regions and land cover types. A simple example would be rainforests and non-desert equatorial regions. Using a strict no-cloud policy would make sampling enough low-resolution images either impossible or would make the temporal difference extremely large (up to 7 years for some AOIs).
> > >
> > > With that in mind, we strived to keep the cloud coverage as low as possible, ideally under 5%, while maintaining the temporal difference as small as possible.
> > >
> > >
> > > > Are you planning to provide pixel-wise cloud mask annotations for mask-guided reconstruction and super-resolution approaches?
> > >
> > > We do provide pixel-wise cloud mask and probability annotations, as well as scene classification masks.
> > >
> > > For the L2A product type, we provide:
> > > - Scene classification layer (SCL) masks, provided by the European Space Agency as described [in this technical guide.](https://sentinels.copernicus.eu/web/sentinel/technical-guides/sentinel-2-msi/level-2a/algorithm).
> > >
> > > For both L1C and L2A, we provide:
> > > - Cloud masks (CLM) and cloud probabilities (CLP) computed using SentinelHub’s s2cloudless, [as described in SentinelHub's documentation.](https://docs.sentinel-hub.com/api/latest/user-guides/cloud-masks/)
> > >
> > >
> > > CLM and CLP are pixel-wise cloud masks and probabilities computed using s2cloudless at a fixed resolution of 160m per pixel.
> > >
> > > SCL is ESA’s pixel-wise scene classification data, based on Sen2Cor processor, computed at 20m, with values for clouds (low probability / unclassified, medium probability, high probability, cirrus).
> > >
> > >
> > > > It would be great to provide code for evaluating on a distinct subset of the dataset exclusively containing cloud-free samples, in order to allow for focusing on the super-resolution task in isolation. Controlling for sample numbers while involving cloudy observations in a separate condition would make for an interesting comparison. This may provide insights about the difficulty added by the cloud-covered observations.
> > >
> > > The code toolbox we provide makes this possible by accepting a CSV file with an explicit list of AOIs to be used. We also provide cloud masks, percentages and scene classification data for each low-resolution revisit, which makes any kind of filtering very straightforward.
> > >
> > > We opted to use all revisits, regardless of their detected cloud coverage, and without any explicitly provided masks, to make the model as applicable in real-world cases as possible. We have found that the model successfuly learns to ignore clouds when using multiple low-resolution inputs.
> > >
> > > We do however appreciate the idea, and while we can't make any promises, we have opened an issue on GitHub to tackle it when the time allows.
> > >
> > >
> > >
> > > ### Score consideration
> > > In light of the discussion of the several points above, would you please kindly consider **whether this submission really is _not even in the lowest half_ of NeurIPS Dataset & Benchmark track papers, as any overall score below 8 would indicate?**   -- see the [Overall score guideline](https://neurips.cc/Conferences/2021/Reviewer-Guidelines) which defines _"8: Top 50% of accepted papers, clear accept"_.

---

> > > > ### Comment · Reviewer_ao84 · 2022-08-28
> > > > **Reviewer Reply**
> > > >
> > > > This is in response to the authors' reply and to comment on selected points:
> > > >
> > > > * * *
> > > >
> > > > > **Sentinel-2 bands:** “We have now addressed this [...] stating that the low-resolution imagery comes in these two product types, containing a different number of bands, [...]”
> > > >
> > > > I’m glad for the clarification, and that products of both preprocessing levels are provided. Having both levels available will be useful for the users when considering other algorithms and pre-trained models---e.g. the s2cloudless algorithm mentioned below, which runs on level 1-C. As part of the broader audience may be unfamiliar with satellite data and their specifics, this clarification will hopefully contribute to minimizing future misunderstandings and making the field more approachable.
> > > >
> > > >
> > > > > **Wording:** “Would you be please be able to point to the precise wordings that you find the most jarring, so we can address those?”
> > > >
> > > > None of the wordings are overly distracting to my mind. However, for the same reason that some (e.g. you, the authors) may prefer the unconventional writing, others (among which are e.g. reviewers) may more or less strongly dislike. I don't subscribe to either and focus on content over style, as long as the latter doesn't get in the way. However, it's my role to at least point out what may be a potential issue to others. Reviewer 33Vk mentioned specific wordings.
> > > >
> > > >
> > > > > **Score consideration:** "[...], if you could consider whether our answers below make you want to amend your scores."
> > > >
> > > > I am very glad for the constructive discussion with the authors and find myself reassured in my rating. On one hand, this implies the authors' discussion with other reviewers and me didn't reveal any substantial issues that might have been overlooked potentially---confirming my verdict of recommending for acceptance. On the other hand, I am thinking my initial rating is appropriate. This is primarily for two reasons:
> > > >
> > > > 1) Following my review, parts of the submission's text were revised and additional explanations provided. While these are valuable, no novel experiments or analysis were part of the revision and an increase of rating should go hand in hand with these. Specifically, my initial comment remarked that the submission's experimental evaluation is kept very brief and this was not substantially resolved in the revision. While the authors find that evaluating further ablations or baselines (such as the proposed [i, ii]) may be beyond the scope of their dataset submission, I wish to disagree. Many other dataset submissions (also being primarily dataset contributions) in this track conduct comparatively more benchmarking. By doing so, this does not only take (part of) the burden of implementing further prior work from future users (thereby increasing the odds of the community adopting the dataset), but may also help in better characterizing the provided data and design choices themselves (e.g. by ablating over sampled land-cover types, percentage of cloud coverage, number of samples etc). While these points can in the future be addressed by the authors or other users, it is a limitation of the current submission that I'd have liked to see (in parts) resolved to justify an increase in rating.
> > > > 2) The ratings $\geq$ 8 imply a relative comparison (such as "Top 50%", "Top 15%"), and in relation to the total of five works I have reviewed for this track and the other submissions I have read, the current rating is well-calibrated and appropriate, to the best of my judgement. This is anecdotally, but this year I have rated only [a single other submission](https://openreview.net/forum?id=bKO6BPtYQA7&noteId=WGKZXmkg2vE&referrer=%5BReviewer%20Console%5D(%2Fgroup%3Fid%3DNeurIPS.cc%2F2022%2FTrack%2FDatasets_and_Benchmarks%2FReviewers%23reviewer-tasks)) higher than yours. While both submissions are convincing on their own and have crucial strengths in common (such as: being appealing to a more general and wider audience, being well-documented and accessible etc), I find it appropriate to reflect in the relative difference of ratings that one work is defining a novel concept while the other is a comparably incremental, albeit very high quality, improvement on prior datasets on a well-established task. Being conceptually novel (in a meaningful manner) is a criterium I wish to see from a certain rating on. How all these publications will turn out and be received by the broader community is very challenging to predict, but I'll keep an eye on works I reviewed in the future and may likewise have to revision my judgement accordingly. This is the most appropriate feasible for now and I wish you all the best for your dataset to become an established standard!
> > > >
> > > >
> > > > I hope this feedback is helpful to you and clarifies my verdict.

---

> > > > > ### Author Response · Authors · 2022-08-29
> > > > > **Thank you for clarification and engaging!**
> > > > >
> > > > > Dear colleague
> > > > >
> > > > > Thank you for taking the time of writing this thorough answer, and providing context.  Very good point on the wording: we cannot depart from the norm and expect everyone to enjoy it, absolutely.
> > > > >
> > > > > We entirely agree on the usefulness of more benchmarking, and with a larger team we would likely have added more methods. This is where it boils down to prioritisation, and having had to decide on where we focused our time and efforts: designing the dataset vs benchmarking it with yet more methods and variations. This is where we decided to focus on novelty.
> > > > >
> > > > > We very much agree with your emphasis on novelty. This is also what the [2022 NeurIPS Main Track reviewer guidelines](https://neurips.cc/Conferences/2022/ReviewerGuidelines)  (which we understand were also due to be used for this track, but for some reason weren't)  also agree, and swap the replace comparison by explicit descriptions for each score -- and indeed 8 and above require "novel ideas".
> > > > >
> > > > > We also agree that this is, by far, not the first satellite imagery dataset, and that, if we limit ourselves to the super-resolution application, there exist other relevant datasets for that task (albeit none, in our opinion but we're biased, as satisfactory). And yes, there are many benchmarks and evaluations.
> > > > >
> > > > > This is why we focused our effort (and the bulk of the paper) on what we believe _is_ novel: the methodology to build this dataset, namely the cross with other carefully assessed reference datasets for multi-source stratified sampling approach with partial class rebalance to maximise feature- and spatial-coverage for a wide range of use cases. This construction itself is applicable for any new dataset for Earth observation (possibly with different choices within the same methodology), and, to the best of our knowledge, no geographic dataset has ever taken such a systematic approach.
> > > > >
> > > > > We believe this is novel, and thus raise it here to make sure it is not being overlooked.
> > > > >
> > > > > This being said, even if you decide not to upgrade your score, we very much appreciate the clarification of your opinion, your feedback, and very much enjoy the discussion!

---

### Official Review · Reviewer_j2VW · 2022-07-25
**Could you please explain the regulation issues?**

**Rating:** 7
**Confidence:** 4
**Correctness:** Please check with lawyers.
**Clarity:** Yes.

**Strengths:**

Your project is very important and I believe it would have a great scientific value.

**Weaknesses:**

However, it may have legal issues.

**Additional Feedback:**

Please check with lawyers.

**Documentation:**

Yes.

**Ethics:**

However, it may have legal issues.

**Relation To Prior Work:**

Yes.

**Summary And Contributions:**

I worked on a Satellite Image project where each image has over 10B pixels. Such images may be illegal to put online.
Your project is very important and I believe it would have a great scientific value.

However, it may have legal issues.  Did you check with lawyers?  In the paper, it mentioned "European Space Agency’s Phi-Lab as part of the ESA-funded QueryPlanet project".  Besides laboratory regulations, there may be national-wide laws. Please check with lawyers.

---

> ### Author Response · Authors · 2022-08-23
> **Answer to reviewer j2VW (7/10, 4/5)**
>
>
> Thank you very much for your review. We address here the specific points you raised, and invite you to also read the [common answer part 1](https://openreview.net/forum?id=DEigo9L8xZA&noteId=CIl-pnhE7Gn) and its [part 2](https://openreview.net/forum?id=DEigo9L8xZA&noteId=xVA9llVJqkX).
>
>
> > However, it may have legal issues. Did you check with lawyers? In the paper, it mentioned "European Space Agency’s Phi-Lab as part of the ESA-funded QueryPlanet project". Besides laboratory regulations, there may be national-wide laws. Please check with lawyers.
>
> Thank you for your vigilance and for the recommendation to check with legal experts if there might be any restrictive national or international laws. Earth Observation is indeed a powerful tool, which comes with strong regulations which vary by country, as you seem very knoweldgeable about.
>
> This is why this project was developed every step of the way in close partnership with the European Space Agency (who funded us), and in particular their Third Party Mission section, in charge of the coordination with providers, including legal review. To further assuage your worries:
> - Every license has been carefully checked. When the default license did not allow for the uses intended (e.g. imagery redistribution), special license was secured with the providers.
> - The high-resolution imagery is published under a properly acquired specific license from Airbus, with written authorization, obtained by the European Space Agency’s Third Party Missions legal team.
> - The partner-provided locations are published with written permission.
> - There is no confidential data included in the dataset.
> - There are no national or international laws preventing the public release of this dataset.
>
> If there are some specific concerns that you fear we might be overlooking in the above list, we would be glad to hear further details -- and if you deem these too sensitive for OpenReview public comments, you are welcome to contact us directly by email.
>
> ### Score consideration
> In light of your clear enthusiasm for this dataset, and hopefully with your concerns having been assuaged, would you please kindly consider **whether this submission really is _not even in the lowest half_ of NeurIPS Dataset & Benchmark track papers, as any overall score below 8 would indicate?**   -- see the [Overall score guideline](https://neurips.cc/Conferences/2021/Reviewer-Guidelines) which defines _"8: Top 50% of accepted papers, clear accept"_.

---

### Official Review · Reviewer_sYLA · 2022-07-25
**An interesting and well considered dataset**

**Rating:** 7
**Confidence:** 3
**Correctness:** The dataset appears to be thoughtfull…
**Clarity:** The paper and datasheet are clear and…

**Strengths:**

- Principled and well-motivated sampling means that models developed against this dataset are likely to be useful in a variety of geographical regimes and use cases. In particular, the combined spatial (under-represented) and semantic (land-use) sampling is of particular interest
- In addition to being well sampled, the dataset is large and allows for the exploration and utilization of high-resolution data to supplement a free data source.
- The dataset is well documented. In particular, the release of exploratory notebooks and code with which to train a model mean that others can begin building on top of this dataset

**Weaknesses:**

- The over-sampling of urban environments is a significant design decision which introduces a bias in the dataset; the motivation for this could be further discussed (since there are plenty of machine learning + remote sensing applications which aren't focussed on urban areas). In particular, by using 'urban' as a proxy for 'settled', the "settlement" portion of the dataset seems to significantly over-sample Europe and North America relative to other parts of the world (Figure 2).

- Related to the above, it would be nice to see some quantification on the geographic distribution of the dataset, since this can influence the applications of the dataset. In addition, worldwide representativity is an explicit goal of this dataset according to the paper.

**Additional Feedback:**

None

**Documentation:**

The data and code are well documented. Storing the data on Kaggle and Zenodo allows multiple methods of accessing the data

**Ethics:**

None known. Potential ethical issues are addressed by the authors in the paper.

**Relation To Prior Work:**

The paper clearly describes prior attempts to solving this problem, and their limitations. In addition, it considers the specific use case that this dataset can solve, framed in previous attempts to solve these problems.

**Summary And Contributions:**

This dataset consists of high resolution Spot 6/7 data (1.5m/pixel resolution), paired with multiple Sentinel-2 revisits (10m/pixel resolution).

This dataset was specifically sampled to a) represent a variety of urban settings and land cover classes, b) cover areas of interest to charities such as Amnesty International.

The stated goal of this dataset (and demonstrated example) is to encourage the development of globally representative super-resolution modelling, allowing Sentinel-2 data to be upsampled to 1.5m resolution.

---

> ### Author Response · Authors · 2022-08-23
> **Answer to reviewer sYLA (7/10, 3/5) (part 1/2)**
>
>
> Thank you very much for your review. We address here the specific points you raised, and invite you to also read the [common answer part 1](https://openreview.net/forum?id=DEigo9L8xZA&noteId=CIl-pnhE7Gn) and its [part 2](https://openreview.net/forum?id=DEigo9L8xZA&noteId=xVA9llVJqkX).
>
>
> > The over-sampling of urban environments is a significant design decision which introduces a bias in the dataset; the motivation for this could be further discussed (since there are plenty of machine learning + remote sensing applications which aren't focussed on urban areas). In particular, by using 'urban' as a proxy for 'settled', the "settlement" portion of the dataset seems to significantly over-sample Europe and North America relative to other parts of the world (Figure 2).
>
> We are very glad that you share our concern about the over-representation of urban environments in existing datasets!
>
> Indeed, at first glance, the class histogram at the bottom of Figure 2 would indicate a massive oversampling of Urban tiles, which, we agree with you, would severely restrict the usability of this dataset.  It is therefore reinsuring to dig into why exactly that proportion seems so high, and indeed what is the difference between "settled" and "urban", and how we avoid the clear biases that such an undifferentiated proxy would entail (and we hope this will convince you it is not undifferentiated here!)
>
> First, the enrichment by "Urban" (in the LCCS sense described below) only concerns 25% of the dataset. Another 25% are all of the non-urban classes in the LCCS, and the remaining 50% come from human rights and refugees focus -- which, as seen in Figure 2, are focused far from Europe and North America.
>
> Second, when it comes to the 25% of the dataset selected by of "Urban", and the risk of confounding with "Settlement", we know the limitations of our expertise and would not pretend to design a classification system ourselves. That is why we use the Land Cover Classification System and as provided by ESA Climage Change Initative (see references for their product user guide in the article), which involved far more knowledgeable geographers than us machine learners. Its class of "Urban" (LCCS) / "Settlement" (IPCC) is the only non-vegetation class, and thus a catch-all. To summarize it in very broad strokes, it roughly captures constructions. It is therefore _not_ "Urban" in the sense of the "a city in Europe or North America". Yet even for these 25% selected as "Urban" in LCCS, we sub-stratify according to the Global Human Settlement Layer "SMOD" classification, which covers from high-density all the way to rural. This further ensures that we do not just sample high-density cities, but indeed cover all types of human densities.
>
> To further confirm the point that "Urban" (in the sense of LCCS and thus of this article) is not as restrictive as feared, and to explain the ungainly peak on "Urban" in the histogram of Figure 2, we observe (not shown in the article due to space limitations) that most of the UNHCR-provided locations for "People of Concern", for example refugee camps, tent villages, etc, also happen to be labelled as "Urban" by the LCCS.
>
> In summary: we absolutely shared your concern from the very beginning of the creation of this dataset, and at each step of the way. We hope that the details of the classification system provided here will assuage your very valid worries! Happy to discuss further if you want.
>
> **(continued in next post due to character limit)**

---

> > ### Author Response · Authors · 2022-08-23
> > **Answer to reviewer sYLA (7/10, 3/5) (part 2/2)**
> >
> > **(continued from previous post due to character limit)**
> >
> > > Related to the above, it would be nice to see some quantification on the geographic distribution of the dataset, since this can influence the applications of the dataset. In addition, worldwide representativity is an explicit goal of this dataset according to the paper.
> >
> >
> > This is an excellent idea, thank you. We provide in Figure 2 two visualisations of the spread of AOIs across the globe, but have not proceeded to formalize it in a table. What would be the ideal grouping for the counts/proportions? By country? By continent?
> >
> > Please keep in mind that we sample by "land use". For example, most of Alaska, northern Canada, and Russian steppes are rich in moss and lichens, so somewhat redundant purely from a land-use point of view, and would thus be under-represented from a geopolitical point of view. Similar concerns would thus inevitably mar all grouping by geopolitical entities.
> >
> > The stratification by LCCS class for 50% of the dataset, and more precisely the uniform geographical sampling amongst revisits _within a given land-use class_, should help mitigate too big a focus on, or imbalance in favour of any particular area -- as opposed to, say, using a dataset built only on the United States.
> >
> > Finally, it is interesting to acknowledge that there will be some bias, actually towards the Global South as opposed to the Global North: the 40% of the dataset centred on UNHCR "People of Concern" dataset seems geographically located more frequently in Africa and the Middle East than in Europe or North America.
> >
> > In summary, we have done the best we could to provide as broad a coverage of the land uses of the world, while still making some difficult judgement calls to accomodate the most likely types of applications. It is a delicate balance, which probably will not satisfy the ideal case for every potential user. But we do hope that, combined with the scale of the dataset, and the ability to extend it, it will provide _some_ value to all these uses.
> >
> > ### Score consideration
> > In light of the discussion of the several points above, would you please kindly consider **whether this submission really is _not even in the lowest half_ of NeurIPS Dataset & Benchmark track papers, as any overall score below 8 would indicate?**   -- see the [Overall score guideline](https://neurips.cc/Conferences/2021/Reviewer-Guidelines) which defines _"8: Top 50% of accepted papers, clear accept"_.

---

### Official Review · Reviewer_Q9Nn · 2022-07-26
**Free High-Resolution Satellite Imagery: The WorldStrat Dataset**

**Rating:** 6
**Confidence:** 3

**Strengths:**

The main strength of this dataset is the availability in an easy to obtain manner and example code to implement super resolution techniques. The authors go to great lengths to address data selection and curation methodology and also address ethical concerns arising from the use of data from some under-served regions of the planet. One application of this dataset is discussed briefly.

**Weaknesses:**

There a few weaknesses in this paper
- The paper uses many acronyms which are not always clearly defined when first used, making the paper difficult to read. For example, having to search the internet for acronyms that may be common knowledge to domain experts or authors themselves makes it harder to go through the paper.
- There is not much discussion of the super resolution application. For instance, the paper discusses three architectures for benchmarking super resolution approaches but does not include any metrics that will help the reader understand the effectiveness of existing approaches. Similarly, there is no mention of actual metrics at all other than referring to another paper.
- It would also be beneficial to highlight how the broader NeurIPS/AI/ML community can leverage this dataset for other applications in machine learning.

**Additional Feedback:**

No additional feedback

**Clarity:**

The liberal use of acronyms throughout the paper made it a little hard to read, specifically because the reader needed to lookup definitions too often. Having clearly defined all acronyms when using them for the first time in the paper would go a long way towards making the paper clearer and easier to understand. At times, the Supplementary Material was actually better organized for understanding the paper. I understand that page limits pose a challenge, but perhaps there's a way to tighten up the structure of the paper and include some additional detail from the supplementary materials. (For instance, removing Figure 5 and using some of that space to discuss the super resolution benchmark comparison metrics)

**Correctness:**

The dataset is constructed appropriately. However, only one potential application is discussed and experiments are not described in great detail. The paper will benefit from a more detailed discussion of the experiments performed and metrics used.

**Documentation:**

Yes, the data and code are easily accessible

**Ethics:**

There is some concern about satellite imagery from conflict area which the authors address.

**Relation To Prior Work:**

Yes, the authors do a good job of discussing prior contributions and how this work differs.

**Summary And Contributions:**

This paper presents a curated high-resolution satellite imagery dataset along with implementations of super resolution using this dataset.

---

> ### Author Response · Authors · 2022-08-23
> **Answer to reviewer Q9Nn (6/10, 3/5)**
>
>
> Thank you very much for your review. We address here the specific points you raised, and invite you to also read the [common answer part 1](https://openreview.net/forum?id=DEigo9L8xZA&noteId=CIl-pnhE7Gn) and its [part 2](https://openreview.net/forum?id=DEigo9L8xZA&noteId=xVA9llVJqkX).
>
>
> > The paper uses many acronyms which are not always clearly defined when first used, making the paper difficult to read.
>
> All acronyms have now been properly introduced, thank you for pointing this out.
>
> > There is not much discussion of the super resolution application. For instance, the paper discusses three architectures for benchmarking super resolution approaches but does not include any metrics that will help the reader understand the effectiveness of existing approaches. Similarly, there is no mention of actual metrics at all other than referring to another paper.
>
> Considering the article mainly targets the "Datasets" portion of the track and the page limitations of the article, an in-depth discussion of the metrics would be detrimental -- especially as they are quite mathematical in nature, common in the super-resolution domain, and very well defined in the referenced paper.
>
> Yet we do agree that the metrics used could be at least summarized, which we have done, along with a more detailed explanation of the training process and the hyperparameters used.
>
> > It would also be beneficial to highlight how the broader NeurIPS/AI/ML community can leverage this dataset for other applications in machine learning.
>
> A very good point, which we have added in the beginning of Section 4.
>
> > The dataset is constructed appropriately. However, only one potential application is discussed and experiments are not described in great detail. The paper will benefit from a more detailed discussion of the experiments performed and metrics used.
>
> We hope the reviewer's concerns have been addressed in the common answer to all reviewers, as this is a point that three reviewers have made.  Keeping in mind that the article primarily targets the "Datasets" part of the track, and the page limitations, we have now added:
>
> - An explanation on how the three models for each of the three architectures were trained,
> - The data splits and number of revisits used to train the models,
> - The optimizers and schedulers used,
> - A summary of the metrics used,
> - A detailed table of all the hyperparameters used in the Appendix of the article.
>
> > The liberal use of acronyms throughout the paper made it a little hard to read, specifically because the reader needed to lookup definitions too often.
>
> We understand that there are a lot of domain-specific acronyms that the reader has to tackle while reading the article. We've now, thanks to you and other reviewers, taken care to properly introduce each acronym, with more important ones being introduced several times to make it as easy as possible for the reader.
>
>  Like you, we wish the geospatial field were lighter on acronyms. There is also no good way around these, since there are a lot of provider, mission and dataset names that are used and have to be introduced to the reader. We hope our edits mitigate this somewhat!
>
>
> ### Score consideration
> In light of the discussion of the several points above, would you please kindly consider **whether this submission really is _not even in the lowest half_ of NeurIPS Dataset & Benchmark track papers, as any overall score below 8 would indicate?**   -- see the [Overall score guideline](https://neurips.cc/Conferences/2021/Reviewer-Guidelines) which defines _"8: Top 50% of accepted papers, clear accept"_.

---

### Official Review · Reviewer_Sf2W · 2022-07-27
**Review of "The WorldStrat Dataset" by Reviewer Sf2W**

**Rating:** 6
**Confidence:** 4
**Correctness:** The claims are almost correct.
**Clarity:** As commented above, the writing needs…

**Strengths:**

1. The proposed dataset covers large area of locations and various types of land-use across the world, which is novel with respect to scale and diversity comparing with existing satellite imagery datasets.
2. The dataset is fully released and can be accessed with ease. The accompanying document and code is also complete.
3. The dataset has potential to benefit research community of both satellite imagery and computer vision.

**Weaknesses:**

1. The writing may need be further refined. Some parts of the main text are a little redundant. It seems like a report with unimportant details rather than a succinct article. On the other hand, details and analysis of the benchmark experiments are not sufficient.
  a. For example in Section 3.3, it is difficult for me to follow the redundant text together with the coarse figure. Figure 4 has inconsistent X-axis and strange number on the grey bar, making it difficult to get meaningful information.
  b. In Section 4.1, the metric of 95% confidence interval (CI) should be explained at least briefly. You can't simply ask the reader to refer to other paper without any explanation.
  c. In Figure 6, the right part looks strange that the three different baselines have exactly the same value on two metrics. The current explanation makes no sense for me. Further discussion is needed.
2. The submission is not formatted in a review mode; there is no line number.

**Additional Feedback:**

Please see my comments above.
In addition, please refine the text carefully.
  a. Please write full name when an abbreviation first apperars, e.g., IPCC and LCCS in Section 2.1.
  b. In Figure 2, it is written 5000 $km^2$ for Underrepresented location, however, there is only 3895 $km^2$ right?
  c. In the beginning of Section 2, what do you mean by The JIF dataset?

**Documentation:**

Documentation is complete as required by the track.

**Ethics:**

NA.

**Relation To Prior Work:**

The difference from related datasets is discussed in Introduction. It may be better to add a seperate section of Related Work to introduce related work of satellite imagery and image super-resolution in a more organized way.

**Summary And Contributions:**

This paper presents a large scale satellite imagery dataset (WorldStrat). The dataset involves high-resolution satellite images covering 10,000 sq km of unique locations of various land uses across the world. Each high-resolution image is paired with multiple low-resolution images of the same location. The dataset can directly facilitate the computer vision task of image super-resolution for satellite imagery. This paper has also conducted benchmark experiments with three compute-efficient baselines.

---

> ### Author Response · Authors · 2022-08-23
> **Answer to reviewer Sf2W (6/10, 4/5)**
>
>
> Thank you very much for your review. We address here the specific points you raised, and invite you to also read the [common answer part 1](https://openreview.net/forum?id=DEigo9L8xZA&noteId=CIl-pnhE7Gn) and its [part 2](https://openreview.net/forum?id=DEigo9L8xZA&noteId=xVA9llVJqkX).
>
> > The writing may need be further refined. Some parts of the main text are a little redundant. It seems like a report with unimportant details rather than a succinct article.
> > Details and analysis of the benchmark experiments are not sufficient.
>
> As discussed in the common response, we have slightly adjusted parts of the writing.
>
> We hope that you will find helpful our emphasis in the common response that the main focus of this article is the dataset. This might explain the imbalance you find between the "unimportant details" of the construction of the dataset, compared to the "not sufficient" details and analysis of the benchmarks and experiments. This is _not_ a Benchmarks paper :)
>
> Nevertheless, keeping in mind that the article primarily targets the "Datasets" part of the track, and the page limitations, we have now added:
>
> - An explanation on how the three models for each of the three architectures were trained,
> - The data splits and number of revisits used to train the models,
> - The optimizers and schedulers used,
> - A summary of the metrics used,
> - A detailed table of all the hyperparameters used in the Appendix of the article.
>
> >    a. For example in Section 3.3, it is difficult for me to follow the redundant text together with the coarse figure.
>
> We are not sure how the text is redundant: could you please clarify what you mean?  If this is that the figure and the main text are repeating each other, we believe that our edits might have now fixed this.
>
> >Figure 4 has inconsistent X-axis and strange number on the grey bar, making it difficult to get meaningful information.
>
> We believe that the X-axis is the same accross all 12 panels of Figure 4. However, we agree that Figure 4 needs to be made clearer for publication -- and commit to doing so for the camera ready version. We have now clarified in the text and in the caption (and will do so in the figure itself) that the "strange number" is the latitude of the Area of Interest, and have made the interpretation clearer in the main text.
>
>
> >    b. In Section 4.1, the metric of 95% confidence interval (CI) should be explained at least briefly.
>
> Thank you for pointing this! We hope the added explanation both in the main text and in the common answer in OpenReview is satisfactory. We have now clarified that the CI is actually a *range* over the validation sample.
>
> >    c. In Figure 6, the right part looks strange that the three different baselines have exactly the same value on two metrics.
>
>
> Yes, absolutely! We are glad that the reviewer thus sees the very behaviour that our 95%-range approach is meant to show: when considering not just the average metric, but the actual distribution accross the validation set, the algorithms are eerily similar! This is further validation for benchmarking the range of the metric rather than its average, as we explain the common answer and, thanks to your input, more clearly in the text now.
>
>
> >    The submission is not formatted in a review mode; there is no line number.
>
> We apologise for the inconvenience. We noticed that the presence or absence of `preprint`  option of the `neurips_data_2022` package provided in the template had quite an impact on pagination this year. This was surprising, as the line numbers in the margin should not take more space. We therefore decided to optimize for what the article would actually look like in its final form, by using that `preprint` option.
>
> > Please write full name when an abbreviation first appears, e.g., IPCC and LCCS in Section 2.1.
>
> All abbreviations have now been properly introduced, thank you for pointing this out.
>
> > b. In Figure 2, it is written 5000 for Underrepresented location, however, there is only 3895 right?
>
> Thank you for pointing this out, there were inconsistencies regarding the total area coverage and they have been fixed.
>
> > c. In the beginning of Section 2, what do you mean by The JIF dataset?
>
> The JIF dataset was an internal working name for the dataset, and it was accidentally left unchanged. This has been fixed, thank you for pointing it out.
>
> ### Score consideration
> In light of the discussion of the several points above, would you please kindly consider **whether this submission really is _not even in the lowest half_ of NeurIPS Dataset & Benchmark track papers, as any overall score below 8 would indicate?**  -- see the [Overall score guideline](https://neurips.cc/Conferences/2021/Reviewer-Guidelines) which defines _"8: Top 50% of accepted papers, clear accept"_.

---

> > ### Comment · Reviewer_Sf2W · 2022-09-02
> > **Vote for acceptance with remaining concern of writing**
> >
> > I have read comments from other reviewers and the authors' relies. I appreciate the authors' effort of collecting a large-scale dataset for super-resolution of satellite imagery. My main concern comes from the writing of the paper, and the reply as well as the revised paper couldn't fully resolve my concern.
> > 1. For the datasets and benchmarks track, the dataset and the benchmark are not isolated. Benchmark results are importance to show how the dataset can be used by other researchers and to verify the usefulness of the dataset. The writing of the two are imbalanced in the paper.
> > 2. Although I am not a native English speaker, I do not like the writing style of the paper from the perspective of acedemic writing. Reviewer-33Vk exactly expresses how I feel about the wording of the paper. Besides, the revised part of the paper is not highlighted and it is not easy to figure out how the paper is updated.
> >
> > Overall, I would like to keep my score.

---

### Author Response · Authors · 2022-08-23
**Common Answer to Reviewers (specific per-reviewer answer follow in respective threads)**

# Common answer to reviewers

We are grateful to our colleagues reviewers for their time and care. As frequent reviewers ourselves, we know that time could be spent on research, and are therefore appreciative of all reviews, even negative -- of which there is remarkably only one versus five positive.

## Overall consensus
Before addressing the very relevant ameliorations suggested, we are humbled by the many enthusiastic comments.

Stating that _"this project is very important"_ (j2VW), _"of great scientific value"_ (j2VW), on a  _"popular topic"_ (ao84), the reviewers highlighted the wide community it can reach, to _"benefit research community of both satellite imagery and computer vision"_ (Sf2W), is _"not only fitting the urging needs of the remote sensing community but also attractive to a broader audience"_ (ao84), and is _"attractive for remote sensing researchers (...) as well as the computer vision community"_ (ao84).

Reviewers also highlighted its size and diversity, as it is _"novel with respect to scale and diversity"_ (Sf2W), _"dataset is large"_ (sYLA), with _"wide and heterogeneous coverage"_ (ao84), _"useful in a variety of geographical regimes and use cases"_ (sYLA).

They kindly lauded its construction: _"thoughtfully constructed, and is well documented."_ (sYLA),  with a _"well-designed geospatial sampling strategy"_ (33Vk), a "Principled and well-motivated sampling"_ (sYLA) ,  _"combined spatial (under-represented) and semantic (land-use) sampling is of particular interest"_ (sYLA).  It overalls _"convinces in its quality of execution"_ (ao84), and even the sole negative review noted that the _"standardised code allows to verify and flexibly extend"_ (33Vk).

Finally, we are thrilled that the reviewers also observed an _"outstandingly great job at making their contribution accessible"_ (ao84), that _"the provided documentation (...) is in great detail and goes beyond the standards of comparable submissions."_ (ao84). They saluted the _"multiple methods of accessing the data"_ (sYLA), _"accessed with ease"_ (Sf2W), _"availability in an easy to obtain manner"_ (Q9Nn), doing a _"great job of documenting their data and making it easily accessible"_ (ao84), with _"execution of curating and documenting (...) outstandingly well done"_ (ao84).

It is always heartening to receive such great feedback as part of the review process.

## Kind request to reviewers

If anything, such laudatary comments make us hopeful, and we respectfully ask the reviewers to consider, in such light: **is this submission really _in the lowest half_ of NeurIPS Dataset & Benchmark track papers, as any overall score below 8 indicates?** -- see the [Overall score guideline](https://neurips.cc/Conferences/2021/Reviewer-Guidelines) which defines _"8: Top 50% of accepted papers, clear accept"_.

As we now detail below how we incorporated their very helpful constructive feedback, we would be grateful if the reviewers would consider the full scoring scale, including and beyond 8/10.

## External validation since submission

The release of this dataset (upon submission of this article) has lead to humbling external validation, and very strong signs of interest in the community:

- [Zenodo already shows **3,929 downloads -- of which 2,251 unique downloads**](https://zenodo.org/record/6810792).
- The IDRIS main compute cluster for [France's CNRS](https://en.wikipedia.org/wiki/French_National_Centre_for_Scientific_Research) has asked to mirror WorldStrat internally for all of CNRS researchers.
- It has been covered in independent blogposts: a researcher not involved in this research, Rob Cole, has written [an entire blog post](https://robmarkcole.com/markdown/2022/08/01/worldstrat.html) lauding the dataset, calling it _significant for a number of reasons_, saying it _raises the bar for documentation quality_ , and being _very impressed by both the size and global nature of the dataset, and the high quality of the documentation and code_, and thinking it _could have significant positive impact on end users_.
- [The GitHub repository has been starred 110 times in a short period of time](https://github.com/worldstrat/worldstrat/stargazers)

**(continued in following post due to character limit)**

---

> ### Author Response · Authors · 2022-08-23
> **(ctd, 2/2)**
>
> **(continued from previous post due to character limit)**
>
> ## Common constructive feedback
>
> A few important points were rightly raised in common by several reviewers.
>
> ### Form: Lack of proper acronym/abbreviation introduction (Sf2W, Q9Nn, 33Vk)
>
> This is a pet peeve of ours too as readers: we absolutely should have defined all acronyms and abbrevations. We blame our final-minute-before-deadline submission. This has now been fixed.
>
> ### Form: Writing style, and figures readability (33Vk, Sf2W, Q9Nn)
>
> The original writing style also partly suffered from pre-submission rush, and we believe we have corrected the most egregious impact. On the broader tone, some of us personally believe that a more conversational style in academic writing can help broaden the readership. However, this is not (yet?) a widespread taste in the community, and we will gladly amend any precise occurence the reviewers would like to point to -- so the form does not distract from the content.
>
> The figures could have been more polished for better readibility, in particular font sizes and axis labels. We are aware of this and in the process now fixing those, which will be fixed by the camera-ready version. We will list in individual answers to each reviewer what we have done for the figures they pointed.
>
> ### Content: Deeper discussion of the benchmark (Sf2W, Q9Nn, ao84)
>
> Three reviewers asked for a deeper discussion of the super-resolution benchmark part in Section 4. We are thrilled that the benchmark part of our article seemed so robust that it lead it to being judged by the same criteria as pure benchmark papers.
>
> We should clarify however that this is a first and foremost a _dataset_ article. This NeurIPS track welcomes dataset articles and benchmark articles, two distinct types of papers. While we believe the super-resolution methods enabled by this dataset are of high interest, we did not set out in this article to provide an _exhaustive_ benchmark of that very active field, which would warrant a completely separate article. As such, our benchmark is here to give a few examples of core methods of the fields, nothing more. The benchmark section serves as a sanity check and provided as a starting point for further research, as an added bonus to a dataset-focused article.
>
> More super-resolution methods are welcome as contributions, which is why we have open-sourced the code with a very permissive license, and are eagerly welcoming pull-requests on the repository.
>
> We have added a brief discussion of the metrics, and elaborated in Section 4 why we report the 95% range of the metrics on the distribution set, rather than limiting ourselves to the more usual 95% confidence intervals. We believe this is a much more rigorous reporting: we report the _95\% range of the metric_ over the whole validation set}, i.e. the 2.5-percentile and the 97.5-percentiles over all data points in the validation sample. It shows the full context of the metric over the whole distribution of inputs. The reader can thus better see whether a few extra points on a metric make a strong difference or are just minor compared to within-distribution variability.
>
>
> We did perform ablation studies several times during the project, but did not include them in the article for the reasons mentioned above. We have now mentioned them in Section 4.1, however without going into too much details as, again, the core focus of this article is not the benchmark.
>
> Finally, we believe this WorldStrat dataset can enable a broad range of applications. While we benchmark here multi-frame superresolution, closest to the authors' own expertise, this is by no means restrictive. More applications can be devised,  either with extra labelling, or using self-supervised representations on low and high resolutions, e.g. extending Tile2Vec (Jean et al. 2019), or even learning transfer tasks from one resolution to the other.

---

### Author Response · Authors · 2022-08-28
**Deadline for discussion tomorrow**

Dear reviewers

We are aware that the deadline for revisions is tomorrow. Apologies for having sent our own answers only 5 days ago -- our bad. We would nevertheless be very glad to engage in conversation, or if time prevents, at least if you could consider whether our answers below make you want to amend your scores.

Thank you for your time.

---

### Meta-Review · Area_Chair_LuGz · 2022-09-03

**Recommendation:** Accept
**Confidence:** 4

**Metareview:**

This paper had 6 reviews. The reviewers’ comments were overall fine, except concerns and the writing style (ao84, Sf2W and 33Vk). Instead of addressing these concerns and showing that they were taken into consideration, the authors responded aggressively, e.g., to reviewer 33Vk.

It has been noted that legal concerns have been addressed in the response to the reviewers’ comments, but in order to prevent any lawsuits against Neurips, I recommend that the authors include a footnote or a specific section in their revised paper to address the legal concerns (in accordance with their response to the reviewers' comments).

In terms of significance, although the work and effort are considerable to construct this dataset, “as it stands, without additional layers for e.g., semantic segmentation, the dataset is most valuable for super-resolution or unsupervised learning approaches” [33Vk]. Maybe future work could include semantic segmentation or add a section to explain how the dataset can be used for other tasks.

Dataset is available on various platforms and documentation seems appropriate.

---

### Decision · Program_Chairs · 2022-09-16

Accept